# PLANT 'N' SEEK:
# CAN YOU FIND THE WINNING TICKET?

**Jonas Fischer**
Max Planck Institute for Informatics
fischer@mpi-inf.mpg.de

**Rebekka Burkholz**
CISPA Helmholtz Center for Information Security
burkholz@cispa.de

## ABSTRACT

The lottery ticket hypothesis has sparked the rapid development of pruning algorithms that perform structure learning by identifying a sparse subnetwork of a large randomly initialized neural network. The existence of such 'winning tickets' has been proven theoretically but at suboptimal sparsity levels. Contemporary pruning algorithms have furthermore been struggling to identify sparse lottery tickets for complex learning tasks. Is this suboptimal sparsity merely an artifact of existence proofs and algorithms or a general limitation of the pruning approach? And, if very sparse tickets exist, are current algorithms able to find them or are further improvements needed to achieve effective network compression? To answer these questions systematically, we derive a framework to plant and hide target architectures within large randomly initialized neural networks. For three common challenges in machine learning, we hand-craft extremely sparse network topologies, plant them in large neural networks, and evaluate state-of-the-art lottery ticket pruning methods. We find that current limitations of pruning algorithms to identify extremely sparse tickets are likely of algorithmic rather than fundamental nature and anticipate that our planting framework will facilitate future developments of efficient pruning algorithms, as we have addressed the issue of missing baselines in the field raised by Frankle et al. (2021). Our code is publicly available at www.github.com/RelationalML/PlantNSeek.

## 1 INTRODUCTION

Deep learning has achieved breakthroughs in multiple challenging areas pertaining to machine learning, in particular in areas for which we lack competitive hand-crafted algorithms. The benefits of overparameterization for training with SGD (Belkin et al., 2019) seem to call for ever wider and deeper neural network (NN) architectures, which are computationally demanding to learn and deploy. Training smaller, adequately regularized NNs from scratch could be a remedy but it commonly seems to fail due to inadequate parameter initialization, as Frankle & Carbin (2019) noted in their seminal paper. As proof of concept that this problem is solvable, they proposed the lottery ticket (LT) hypothesis, which states that a small, well trainable subnetwork can be identified by pruning a large, randomly initialized NN, opening the field to discover such subnetworks or 'winning tickets'.

Based on the findings of Zhou et al. (2019), Ramanujan et al. (2020) went even further and conjectured the existence of strong lottery tickets, i.e., subnetworks of randomly initialized NNs that do not require any further training. This strong LT hypothesis holds the promise that training NNs could potentially be replaced by efficient NN pruning, which simultaneously performs structure learning by identifying a task specific sparse neural network architecture. The existence of strong LTs has also been proven formally for networks without (Malach et al., 2020; Pensia et al., 2020; Orseau et al., 2020) and with potentially nonzero biases (Fischer et al., 2021; Burkholz et al., 2022).

While these types of proofs show existence in realistic settings, the sparsity of the constructed tickets is likely not optimal, as they represent a target parameter by multiple neurons of degree 1. The construction and proof of the generalized strong LT hypothesis raises two question – is the suboptimal sparsity merely an artifact of existence proofs or a general limitation of the pruning approach? And, if very sparse tickets exist, are current algorithms able to find them or are further improvements needed to achieve effective network compression? These questions cannot be answered by compar-

ing LT pruning algorithms solely on standard benchmark datasets (Frankle et al., 2021), but demand the comparison with known ground truth LTs. To fill this gap and generate baselines with known ground truth, we here propose an algorithm to plant and hide arbitrary winning tickets in randomly initialized NNs and construct sparse tickets that reflect common challenges in machine learning. We use this experimental set-up to compare state-of-the-art pruning algorithms designed to search for lottery tickets.

Our results indicate that state-of-the-art methods achieve only sub-optimal sparsity levels. This suggests that previous challenges to identify highly sparse winning tickets as subnetworks of randomly initialized dense networks (Frankle et al., 2020; Ramanujan et al., 2020) can be explained by algorithmic limitations rather than fundamental problems with LT existence. In our experiments, the qualitative trends how methods compare to each other are consistent with previous results on image classification tasks (Tanaka et al., 2020; Frankle et al., 2021) indicating that our experimental set-up exposes pruning algorithms to realistic challenges. In addition, we identify an opportunity to improve state-of-the-art pruning algorithms in order to find strong LTs of better sparsity. Our proposed planting framework will enable the evaluation of future progress in this direction.

**Contributions** 1) We prove the existence of strong lottery tickets with sparse representations. 2) Inspired by the proof, we derive a framework that allows us to plant and hide strong tickets in neural networks and thus create benchmark data with known ground truth. 3) We construct sparse representations of four types of tickets that reflect typical machine learning problems. 4) We systematically evaluate state-of-the-art pruning methods that aim to discover tickets on these three problems against the ground truth tickets and highlight key challenges.

## 1.1 RELATED WORK

LT pruning approaches for neural networks can be broadly categorized into three groups, pruning before, during, or after training. While methods that sparsify the network during (LeCun et al., 1990; Mozer & Smolensky, 1989; Han et al., 2015; Frankle & Carbin, 2019; Srinivas & Babu, 2016; Lee et al., 2020), or after training (Savarese et al., 2020; LeCun et al., 1990; Hassibi & Stork, 1992; Dong et al., 2017; Li et al., 2017; Molchanov et al., 2017) help in reducing computational resources required for inference, they are, however, less helpful in reducing resources at training time but can make a difference if they prune early aggressively (You et al., 2020). They are most useful for structure learning at lower sparsity levels (Su et al., 2020; Lee et al., 2020).
The LT hypothesis (Frankle & Carbin, 2019) has also promoted the development of neural network pruning algorithms that prune *before* training (Wang et al., 2020; Lee et al., 2019; Verdenius et al., 2020; Tanaka et al., 2020; Ramanujan et al., 2020). Usually, these methods try to find LTs in a 'weak' (but powerful) sense, that is to identify a sparse neural network architecture that is well trainable starting from its initial parameters. These methods score edges in terms of network flow, which can be quantified by gradients at different stages of pruning, or based on edge weights, and prune all edges with the lowest scores until the desired sparsity is achieved (Frankle et al., 2021).
Strong LTs are sparse sub-networks that perform well with the initial parameters, hence do not need to be trained any further (Zhou et al., 2019; Ramanujan et al., 2020). Their existence has been proven by providing lower bounds on the width of the large, randomly initialized neural network that contains them (Malach et al., 2020; Pensia et al., 2020; Orseau et al., 2020; Fischer et al., 2021; Burkholz et al., 2022). In addition, it was shown that multiple candidate tickets exist that are also robust to parameter quantization (Diffenderfer & Kailkhura, 2021).

Beyond LT pruning, many more methods have been developed to reduce computational resources and perform structure learning, including dynamic sparse training (Evci et al., 2020; Liu et al., 2021b), adaptations (Frankle et al., 2020; Renda et al., 2020; Liu et al., 2021a) of Iterative Magnitude Pruning (IMP) Han et al. (2015); Frankle & Carbin (2019) and sparse regularization techniques (Weigend et al., 1991; Savarese et al., 2020). As these approaches do not identify LTs as subnetworks of randomly initialized NNs, they do not rely on the existence of planted tickets and are therefore beyond the scope of our experimental analysis. However, the ground truth tickets which we derived for planting could still provide an interesting baseline to explore whether sparse training of deep NNs can identify extremely sparse, hand designed NN architectures. Dense NNs are known to find NN representations that are less sparse than hand crafted architectures (Denker et al., 1987), yet, the explicit objective of sparse training is to address this issue. We provide the tools to evaluate progress in this direction by planting known ticket architectures. While the ultimate goal of deep

learning is to solve problems with otherwise unknown solutions like image classification (Frankle et al., 2021) or protein structure prediction (Tunyasuvunakool et al., 2021), the design of NN architectures for human solvable problems has already in the past provided important insights into NN properties, including universal approximation (Scarselli & Tsoi, 1998; Yarotsky, 2018) or the importance of algorithmic alignment (Xu et al., 2020). NNs that compute polynomials (Scarselli & Tsoi, 1998; Yarotsky, 2018), xor gates (Rumelhart et al., 1986), discrete fast fourier transformation (Velik, 2008), symmetry groups (Sejnowski et al., 1986), general piecewise linear functions (Arora et al., 2018), or argmax (Xu et al., 2020) could also present interesting candidates for planting in future investigations.

## 1.2 NOTATION AND TERMINOLOGY

Let $f(x)$ denote a bounded function, without loss of generality $f : [-1,1]^{n_0} \to [-1,1]^{n_L}$, that is parameterized as a deep neural network with architecture $\bar{n} = [n_0, n_1, ..., n_L]$, i.e., depth $L$ and widths $n_l$ for layers $l = 0, ..., L$ with ReLU activation function $\phi(x) := \max(x, 0)$. It maps an input vector $\boldsymbol{x}^{(0)}$ to neurons $x_i^{(l)}$ as $\boldsymbol{x}^{(l)} = \phi\left(\boldsymbol{W}^{(l)}\boldsymbol{x}^{(l-1)} + \boldsymbol{b}^{(l)}\right)$, where $\boldsymbol{W}^{(l)} \in \mathbb{R}^{n_{l-1} \times n_l}$ is the weight matrix, and $\boldsymbol{b}^{(l)} \in \mathbb{R}^{n_l}$ is the bias vector of Layer $l$. We will establish approximation results with respect to the supremum norm $\|g\|_\infty := \sup_{x \in [-1,1]^{n_0}} \|g\|_2$ defined for any function $g$ on the domain $[-1,1]^{n_0}$. Assume furthermore that a LT $f_\epsilon$ can be obtained by pruning a large mother network $f$, which we indicate by writing $f_\epsilon \subset f_0$. If $f_\epsilon$ achieves a similar performance after training the non-zero parameters as training all parameters of $f_0$, we call $f_\epsilon$ a *weak LT*. If $f_\epsilon$ does not require any further training, we call it a *strong LT*. Note that every strong LT is automatically also a weak LT and any algorithm that prunes the initial network $f_0$ identifies a LT that could, in principle, be a strong LT. The sparsity level $\rho$ of $f_\epsilon$ is then defined as the fraction of non-zero weights that remain after pruning, i.e., $\rho = \left(\sum_l \left\|\boldsymbol{W}_\epsilon^{(l)}\right\|_0\right) / \left(\sum_l \left\|\boldsymbol{W}_0^{(l)}\right\|_0\right)$, where $\|\cdot\|_0$ denotes the $l_0$-norm, which counts the number of non-zero elements in a vector or matrix. Another important quantity that influences the existence probability of lottery tickets is the in-degree of a node $i$ in layer $l$ of the target $f$, which we define as the number of non-zero connections of a neuron to the previous layer plus 1 if the bias is non-zero, i.e., $k_i^{(l)} := \left\|\boldsymbol{W}_{i,:}^{(l)}\right\|_0 + \left\|b_i^{(l)}\right\|_0$, where $\boldsymbol{W}_{i,:}^{(l)}$ is the $i$-th row of $\boldsymbol{W}^{(l)}$. The maximum degree of all neurons in layer $l$ is denoted as $k_{l,\max}$.

## 2 EXISTENCE OF STRONG LOTTERY TICKETS

Pruning algorithms that search for strong LTs achieve sparsity levels of around 0.5 but not substantially smaller if the resulting models should be able to compete with the accuracy of the entire, trained mother network (Ramanujan et al., 2020). Proofs of the existence of strong lottery tickets give no clear indication whether this is an algorithmic shortcoming, which could be overcome, or a fundamental limitation of pruning randomly initialized networks alone. The reason is that existing proofs (Malach et al., 2020; Pensia et al., 2020; Orseau et al., 2020; Fischer et al., 2021) guarantee high existence probabilities of subnetworks that have double the depth and $2 - 30$ times the width of the target network and thus non-optimal sparsity. Based on their $2L$ construction, Malach et al. (2020) even went so far to conclude that training by pruning might be computationally at least as hard as training shallower NNs. However, it is well known that specific function classes can be approximated in significantly more parameter efficient ways by deeper NNs rather than shallower ones (Mhaskar et al., 2017; Yarotsky, 2018) and also be learned more efficiently (Schmidt-Hieber, 2020). Thus, by leveraging its full depth, the randomly initialized $2L$ deep NN might contain a much sparser strong LT than any of the ones whose existence has been proven.

As a first step towards making claims about the existence of very sparse representations, we therefore prove next a lower bound on the probability that a target NN of general architecture is contained in a larger, randomly initialized NN with the same depth as the target network. As many relevant targets have known representations of lower sparsity than what is covered by this bound, we will afterwards propose a planting algorithm to design experiments that can distinguish between algorithmic and fundamental limitations of pruning for strong LTs.

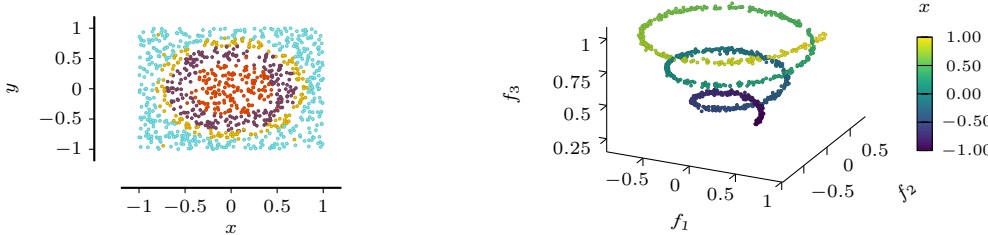

Figure 1: *Benchmark data.* Shown are samples from the `Circle` (left) and `Helix` (right) task.

### 2.1 LOWER BOUND ON EXISTENCE PROBABILITY

Pruning a randomly initialized NN usually finds a strong LT that is close to a target network but does not recover the original parameters exactly. First, we need to understand how these errors in the parameters affect the final network output and what error sizes are acceptable. For completeness, we restate Lemma 1 of Fischer et al. (2021) that guarantees an $\epsilon$ approximation of the entire network.

**Lemma 1** (Error propagation). *Assume $\epsilon > 0$ and let the target network $f$ and its approximation $f_\epsilon$ have the same architecture. If every parameter $\theta$ of $f$ and corresponding $\theta_\epsilon$ of $f_\epsilon$ in layer $l$ fulfils $|\theta_\epsilon - \theta| \leq \epsilon_l$ for*

$$\epsilon_l := \epsilon \left( L\sqrt{n_l k_{l,max}} \left( 1 + \sup_{x \in [-1,1]^{n_0}} \left\| \boldsymbol{x}^{(l)} \right\|_1 \right) \prod_{k=l+1}^{L} \left( \left\| \boldsymbol{W}^{(l)} \right\|_\infty + \epsilon/L \right) \right)^{-1},$$

*then it follows that $\|f - f_\epsilon\|_\infty \leq \epsilon$.*

Respecting the allowed errors $\epsilon_l$, we can next establish a lower bound on the existence probability of a specific target network assuming standard initialization schemes with necessary non-zero bias initialization (Fischer et al., 2021). The main argument is a union bound over matching each target neuron $i$ (with $k_i$ parameters) with neurons of the mother network in the corresponding layer.

**Theorem 2** (Lower bound on existence probability). *Assume that $\epsilon \in (0,1)$ and a target network $f$ with depth $L$ and architecture $\bar{n}$ are given. Each parameter of the larger deep neural network $f_0$ with depth $L$ and architecture $\bar{n}_0$ is initialized independently, uniformly at random with $w_{ij}^{(l)} \sim U\left(\left[-\sigma_w^{(l)}, \sigma_w^{(l)}\right]\right)$ and $b_i^{(l)} \sim U\left(\left[-\prod_{k=1}^{l} \sigma_w^{(k)}, \prod_{k=1}^{l} \sigma_w^{(k)}\right]\right)$. Then, $f_0$ contains a rescaled approximation $f_\epsilon$ of $f$ with probability at least*

$$\mathbb{P}\left( \exists f_\epsilon \subset f_0 : \|f - \lambda f_\epsilon\|_\infty \leq \epsilon \right) \geq \prod_{l=1}^{L} \left( 1 - \sum_{i=1}^{n_l} (1 - \epsilon_l^{k_i})^{n_{l,0}} \right),$$

*where $\epsilon_l$ is defined as in Eq. (1) and the scaling factor is given by $\lambda = \prod_{l=1}^{L} 1/\sigma_w^{(l)}$.*

We could obtain similar results for initially normally distributed weights and biases, we would just have to substitute $\epsilon_l$ by $\epsilon_l/2$. A proof is provided in Appendix A.2.

Thm. 2 provides us with an intuition for what kind of targets we can expect to find. First of all, it tells us that a large number of nodes in a layer, and more importantly nodes with large in-degree $k_i$, render the existence of a specific network architecture as strong LT less likely. Each additional layer reduces the probability further. Moreover, we observe that the last layer is a bottleneck, as it usually has the same width as in the large initial network. A higher width of the mother network is clearly advantageous. Note that we could turn this theorem also into a lower bound on the width $n_{l,0}$ of the larger mother network as it is common in existence proofs. Fig. 6 in Appendix A.2 supports this intuition with the visualization of an example. Assuming the same width $n_{l,0} = n_0$ and $n_l = n$ across layers, we would receive roughly $n_0 \geq C \log(Ln/\delta) \max_l \left( \epsilon_l^{-k_{max}} \right)$. Even though it is polynomial in the relevant parameters, it only provides a practical existence proof for extremely sparse architectures. We therefore have to resort to planting to answer fundamental questions about abilities of pruning algorithms. In fact, the proof of the above theorem inspires the planting algorithm introduced next.

## 2.2 PLANTING STRONG LOTTERY TICKETS

As we have discussed, the LTs that exist with high probability rarely fulfill criteria of interest, such as low sparsity, favorable generalization properties, or adversarial robustness. We therefore propose to plant winning tickets with such desirable properties within randomly initialized neural networks. This approach offers the flexibility to design experiments of different degrees of difficulty and generate training and test data based on a ground truth.

A simple approach to planting a target $f$ in a network $f_0$ would be to select a random subset of neurons in each layer and set them to their target values and otherwise randomly initialize the rest. This, however, would usually lead to a trivially detectable ticket because the target parameters are much larger than the initialized parameters of the larger mother network. The reason is that both networks produce output that lies in a similar range (ideally the one of the training labels). Yet, the target network has to achieve this by adding up a much smaller number of parameters. A different perspective on the same issue is that a pruned lottery ticket needs to be scaled up to compensate for the lost parameters. Note the scaling factor $\lambda$ in Theorem 2 for that purpose. Thus, at least, we would need to scale the target parameters appropriately during planting.

We follow a more general approach that also applies to networks $f_0$ whose parameters have not been randomly initialized and that captures the full variability of possible target solutions by allowing for different scaling factors per neuron. We search in each layer of $f_0$ for suitable neurons that best match a target neuron in $f$, starting from the first layer. Given that we matched all neurons in layer $l-1$, we try to establish their connections to neurons in the current layer $l$. A best match is decided by minimizing the l2-distance to its potential input parameters thereby adjusting for an optimal scaling factor. For example, let neuron $i$ in Layer $l$ of the target $f$ have non-zero parameters $\boldsymbol{\theta} = (b, \boldsymbol{w})$ that point to already matched neurons in Layer $l-1$. Each neuron $j$ in Layer $l$ of $f_0$ that we have not matched yet could be a potential match for $i$. Let the corresponding parameters of $j$ be $\boldsymbol{m}$. The match quality between $i$ and $j$ is assessed by $q_\theta(m) = \|\boldsymbol{\theta} - \lambda(m)\boldsymbol{m}\|_2$, where $\lambda(m) = \boldsymbol{\theta}^T \boldsymbol{m} / \|\boldsymbol{m}\|_2^2$ is the optimal scaling factor. The best matching parameters $m^* = \arg\min_m q_\theta(m)$ are replaced by rescaled target parameters $\theta/\lambda(m^*)$ in $f_0$. We provide pseudocode and details in App. A.3.

## 2.3 CONSTRUCTION OF TARGETS FOR PLANTING

Based on the proposed planting algorithm, we generate sparse tickets for three problems that expose general pruning algorithms to common challenges in machine learning: a basic classification problem, regression problem, and manifold learning problem (Bishop, 2006). On purpose, these are designed to avoid high computational burdens and, most importantly, have sparse neural network architectures with variable depth.

**Regression of a ReLU unit** (`ReLU`) The ReLU unit $\phi(x)$ is an essential building block of state-of-the-art neural networks. It is particularly interesting to study because we know the optimal solution and can guarantee that it exists with high probability. Assuming a mother network $f_0$ of depth $L$, a ReLU can be implemented with a single neuron per layer. Any path through the network with positive weights $\prod_{l=1}^L \phi(w_{i_{l-1}i_l}x)$ defines a ReLU with scaling factor $\lambda = \prod_{l=1}^L w_{i_{l-1}i_l}$ for indices $i_l$ in Layer $l$ with $w_{i_{l-1}i_l} > 0$. Note that each random path fulfills this criterion with probability $0.5^L$ so that even random pruning has a considerable chance to find an optimal ticket. A winning path exists with probability $\prod_{l=1}^L (1 - 0.5^{n_{l,o}})$, which is almost 1 even in relatively small networks. As a ticket with optimal sparsity exists with high probability, we would not need to plant it. However, to give also pruning algorithms a chance that do not specifically prune biases, we set ticket biases to zero. As we will see in experiments, despite this simplification, pruning algorithms are severely challenged and cannot find an optimally sparse ticket.

**Classification of rings** (`Circle`). Another basic building block of many functions, in particular, radial symmetric functions, is the radius or l2-norm of a vector. It is also an important operation to represent products via the relation $xy = 0.25((x+y)^2 - (x-y)^2)$. We derived a sparse representation that leverages the full depth of a given network, as its sparsity improves with increasing depth. In Fig. 1 we visualize the related 4-class classification problem with 2-dimensional inputs. The output is 4-dimensional, where each output unit $f_c(x)$ corresponds to the probability $f_c(x)$ that an input $(x_1, x_2) \in [-1, 1]^2$ belongs to class $c$ that is computed with softmax activation functions. The decision boundaries are defined in the last layer based on inputs of the form $g(x_1, x_2) = x_1^2 + x_2^2$. The

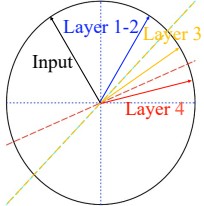 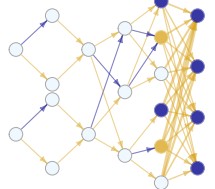 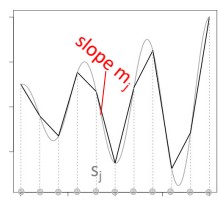 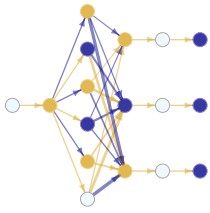

(a) Mirroring along axes for `Circle`.

(b) `Circle` architecture with depth $L = 5$.

(c) Univariate function approximation.

(d) `Helix` architecture with depth $L = 5$.

Figure 2: (a) Visualization of the first layers of `Circle` representing $g(x_1, x_2) = x_1^2 + x_2^2$. (c) Univariate deep neural network parametrization with outer weights $a_j^{(i)} = \Delta m_j = m_j - m_{j-1}$. (b+d) Ticket architectures, edge width is proportional to the absolute weight value, blue indicates a negative sign, yellow a positive sign. Neurons are colored by bias sign, gray indicates zero biases.

high symmetry of $g(x_1, x_2)$ allows us to construct a particularly sparse representation by mirroring data points along axes as visualized in Figure 2 (a). Each consecutive layer $l$ mirrors the previous layer along the axis $\boldsymbol{a}^{(l)} = (\cos(\pi/2^{l-1}), \sin(\pi/2^{l-1}))$. To enable higher precision for networks of smaller depth, the second to last layer approximates $h(x) = x^2$ for each component. This is unnecessary for representations of high enough depth. The details of our construction are explained in App. A.4.2. Fig. 2 (b) shows an exemplary architecture of the planted ticket, for which we can vary the depth and width of the second to last layer. The lower bound on the existence probability given by Thm. 2 (without planting) for the architecture shown in Fig. 2 (b) is numerically 0 for mother networks of width $n_l = 100$ as in our experiments and $\epsilon_l = 0.005$, even when we disregard the last bottleneck layer. For unrealistic high width $n_l = 10^5$ however, the bound is $0.47$ and thus detection becomes theoretically feasible for extreme sparsity levels.

**Identification of a submanifold** (`Helix`) Another common problem in machine learning is to learn lower dimensional functions that are embedded in a higher dimensional space. In Fig. 1 we show our minimal regression example in form of a helix. As we have observed that many pruning algorithms have the tendency to keep a higher number of neurons closer to the input (and sometimes also the output layer), we construct a ticket that has similar properties, see Fig. 2 (d). This should ease the task for pruning algorithms to find the planted winning ticket but, as we show in our experiments, this `Helix` problem is surprisingly challenging. The helix has three output coordinates $f_1(x) = (5\pi + 3\pi x) * \cos(5\pi + 3\pi x)/(8\pi)$, $f_2(x) = (5\pi + 3\pi x) * \sin(5\pi + 3\pi x)/(8\pi)$, and $f_3(x) = (5\pi + 3\pi x)/(8\pi)$ for 1-dimensional input $x \in [-1, 1]$. We can approximate each of the components $f_i(x)$ by an univariate deep neural network $n_i(x) = \sum_{j=1}^N a_j^{(i)} \phi(x - s_j) + b^{(i)}$ with depth $L = 2$ (see Fig. 2 (c) and App. A.4.3), which is achieved by the first layers, while the last layers basically represent the identity. An interesting feature of this example is that highest width could be assigned to almost any layer of the architecture, which makes `Helix` a good candidate to provoke layer collapse in different pruning algorithms. The lower bound on the existence probability given by Thm. 2 (without planting) for the architecture shown in Fig. 2 (d) is numerically 0 for mother networks of any reasonable width, because nodes in the third layer have too large in-degrees. We therefore have to resort to planting.

**Strong tickets based on trained neural networks** Even though we cannot expect to construct sparse baseline solutions for benchmark image classification tasks, we can leverage the fact that weak LTs can currently be identified at lower sparsity levels than strong LTs (see our experiments). To answer the question whether state-of-the-art pruning algorithms can find sparse strong LTs in the setting of standard benchmark data, we plant a trained weak LT in a randomly initialized (VGG like) neural network. Note that the proposed pruning algorithm can also be applied to convolutional layers in addition to fully connected ones.

## 3 EXPERIMENTS

We utilize our planting framework to answer the question whether LT pruning algorithms that identify subnetworks of randomly initialized neural networks are able to identify highly sparse LTs,

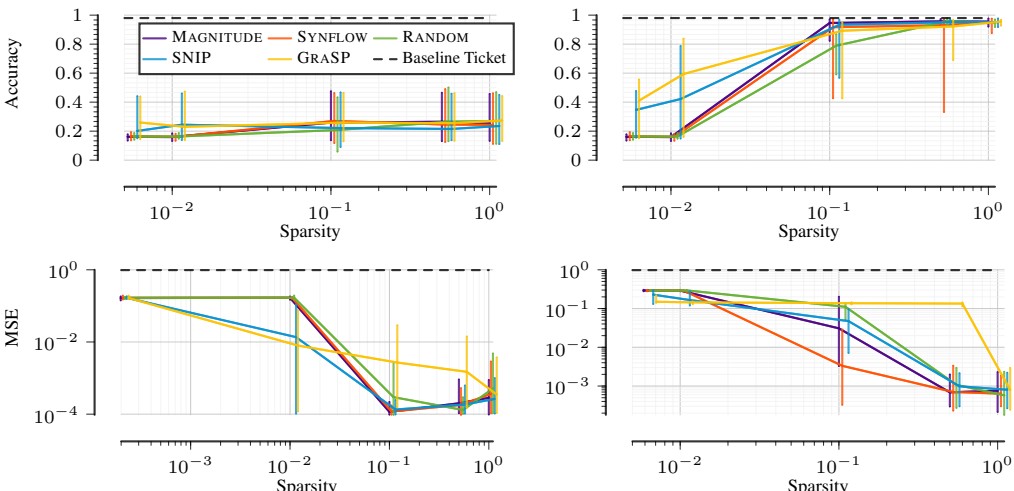

Figure 3: *Singleshot results.* Performance of discovered tickets against target sparsities as mean and value ranges (minimum and maximum) across 25 runs. In order of appearance: `Circle` after pruning (strong ticket), `Circle`, `ReLU`, and `Helix` after training (weak ticket). Results after pruning look similar across tasks. Baseline ticket (leftmost sparsity) is given by black dashed line.

ideally in a strong sense but we also analyze weak LTs. Hypothetically, it could be possible that pruning algorithms for weak LTs only have to resort to training the identified LT because a highly sparse strong LT does not exist with high probability. Yet, if we guarantee the existence of a sparse strong LT, the algorithms would be able to find it and would not require further training. Similarly, if we insist on finding extremely sparse architectures, it might be necessary to give up the search for initial LTs (Frankle et al., 2020; Renda et al., 2020; Liu et al., 2021a). If this were true, we should be able to find highly sparse LTs with the original pruning algorithm if we ensure the existence of a solution by planting.

We reject these hypotheses with our experiments, in which we randomly initialize a dense neural network of width $n_l = 100$ and depth $L = 5$ by He initialization with nonzero biases (Fischer et al., 2021) and plant one of our constructed targets into the initial network. To show that our experiments reflect realistic conditions, we also compare the general trends to results on standard image classification. For reproducibility, we provide details on data, networks, and experimental setup in App. B.1. In general, we report averages over results across 10 independent runs, as well as obtained value ranges as minimum and maximum. We compare only pruning methods that identify lottery tickets as subnetworks of randomly initialized neural networks, as these could potentially find our planted solution or an equally performing one. GRASP, SNIP, SYNFLOW, MAGNITUDE pruning, and RANDOM pruning (Wang et al., 2020; Lee et al., 2019; Tanaka et al., 2020; Frankle & Carbin, 2019) are thus considered, which are algorithms to discover weak tickets, and EDGE-POPUP, which is designed to find strong tickets (Ramanujan et al., 2020).

We distinguish two different pruning approaches, singleshot (see Fig. 3) and multishot (see Fig. 4). In singleshot pruning, which is originally applied in SNIP, SYNFLOW, and GRASP, edges are scored in a single pass and then pruned to the desired sparsity. Compared to multishot pruning, this saves a significant amount of resources by preventing training entirely if a strong ticket is found. If a weak ticket is found, only a small subnetwork needs to be trained once. Multishot pruning leads usualy to better results (Frankle et al., 2021), because it relies on updated gradient information. Analogous to iterative magnitude pruning, for each round $r$, we iteratively reduce the sparsity to $\rho^{r/10}$, where $\rho$ is the desired network sparsity. Within each round, the current subnetwork is first trained, then pruned to the current target sparsity, and then reset to initial parameters for the next round. We analyze the performance of tickets before training to assess whether they qualify as strong LTs and after training to evaluate whether at least pruning for weak LTs is feasible and can identify LT of sparsities that can compete with our planted ground truth.

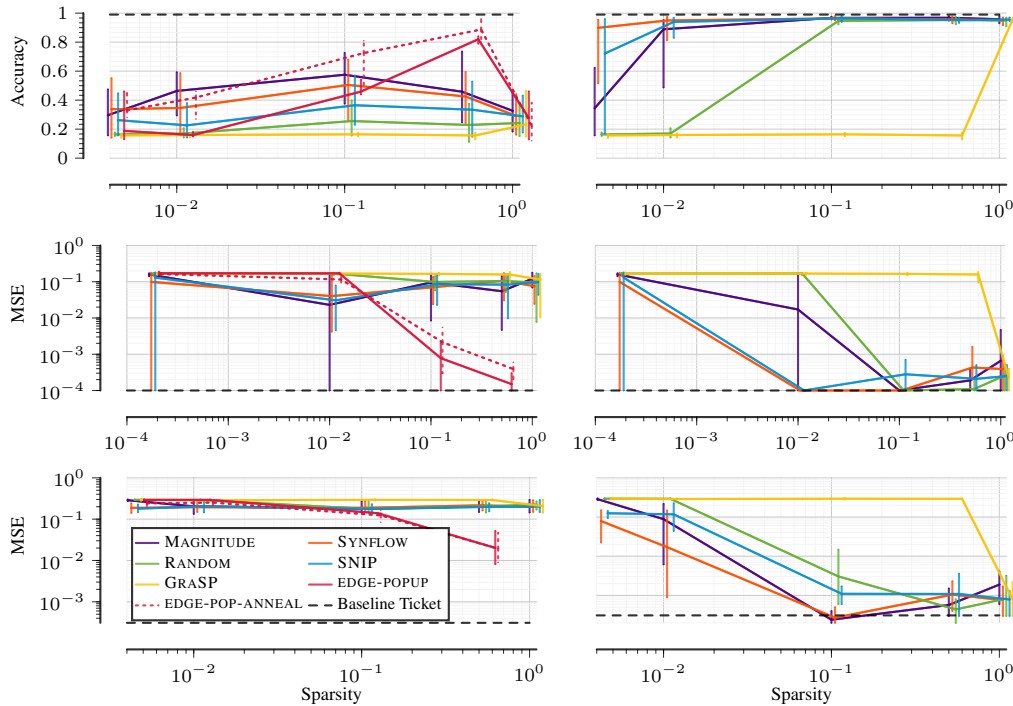

Figure 4: *Multishot results.* Performance for `Circle` (top), `ReLU` (middle), and `Helix` (bottom) for 10 rounds of alternating pruning and training. We provide mean and obtained intervals (minimum and maximum) of accuracies of the final pruned network across 10 repetitions before (strong ticket, left) and after (weak ticket, right) final training. Baseline ticket accuracy is indicated in black.

**Hand designed ground truth** First, we note that training the full network (at sparsity level 1.0) can solve each of our tasks. Thus, planting does not destroy the general trainability of the initial mother network. In fact, we would observe the same performances without planting. Second, we find that none of the approaches is able to discover strong tickets, in particular not the planted ticket, in a single shot. Moreover, only at sparsity levels $\geq 0.1$ the methods are able to find weak tickets. One of the problems that arise is layer collapse. While layer-wise pruning, i.e., setting a target sparsity per layer, would prevent this collapse, it still results in an interruption of flow. For `Circle` classification task, certain multishot methods are able to recover weak tickets, even at extreme sparsities with just marginal performance decrease compared to the ground truth ticket. However, only EDGE-POPUP is able to recover strong tickets for this task, and those tickets have only sparsity levels orders of magnitude larger than the planted baseline even though we were able to improve the original EDGE-POPUP algorithm using the described iterative pruning scheme. Furthermore, we observe that all considered methods struggle more with regression tasks rather than the classification task, as they can only recover weak tickets of sparsity 0.1 or 0.01 and no strong tickets.

A more detailed analysis for varied network depth, width, noise levels, and pruning strategies, alongside a comparison with results on standard benchmark data is presented in the appendix. In summary, we observe similar trends for varying depth and width and find that the analyzed pruning algorithms seem to be robust to noise in the data. Furthermore, how the methods compare relative to each other is in line with experiments on image data as reported in the literature (Tanaka et al., 2020; Frankle et al., 2021). In particular, for all these methods a similar drop around 0.01 sparsity is observed for different VGG and Resnet architectures. This suggests that our simplified learning tasks expose pruning algorithms to challenging and in many ways realistic conditions.

**VGG with strong tickets** While no ground truth solution is known for image classification tasks on standard benchmark datasets, we can still use our planting framework to answer meaningful questions in this context, as we demonstrate next. To test the hypothesis whether EDGE-POPUP is limited to discover strong tickets of suboptimal sparsity of around 0.5, we investigate its capabilities

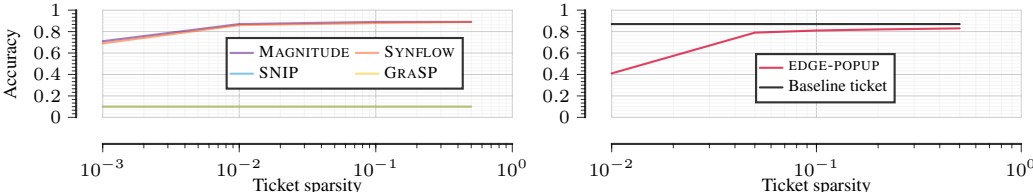

Figure 5: *VGG16 CIFAR10 results.* (Left) Performance for learned weak tickets. (Right) Performance of strong tickets discovered by EDGE-POPUP for VGG with planted baseline ticket of sparsity 0.01. Baseline ticket performance is indicated by black line.

to recover a planted baseline ticket from VGG16. For that, we use SYNFLOW to discover a weak ticket of sparsity 0.01 from VGG16 with multishot pruning, train the weak ticket on CIFAR10, and plant it back into the network. Running EDGE-POPUP on this network, we observe that it indeed cannot retrieve the baseline ticket of desired sparsity in this real world setting (see Fig. 5 right).

**Comparison with ground truth** Our planting framework is designed for the analysis of LT algorithms that seek for subnetworks of randomly initialized NNs. Structure learning methods can also follow a different approach. While planting is less relevant in this case, we can still compare their results with our hand-crafted solutions. We show in App. B.4 that neither dynamic sparse training (Evci et al., 2020) nor LT fine-tuning techniques (Liu et al., 2021a; Renda et al., 2020) find architectures that are competitive with our constructed ground truth tickets.

## 4 DISCUSSION & CONCLUSION

We investigated the optimality of existing lottery ticket (LT) pruning methods and their potential for improvement, both regarding the discovery of strong tickets – subnetworks that perform well at initialization, as well as weak tickets – subnetworks that perform well after training. Recent works, in particular by Frankle et al. (2021), evaluated LT pruning methods and showed that no single best method across considered settings and sparsities exists, and raised the issue of missing baselines in the field. To tackle this issue, we here proposed an algorithm that plants and hides target networks within a larger network, thus allowing to generate baseline tickets for rigorous benchmarking. For three common challenges in machine learning, a classification, regression, and manifold learning problem, we hand-crafted extremely sparse network topologies, planted them in large, randomly initialized neural networks, and evaluated the state-of-the-art pruning methods in combination with different pruning strategies.

Our results indicate that state-of-the-art LT pruning methods achieve in general sub-optimal sparsity levels, and are not able to recover LTs that are competitive with a planted ground truth. This suggests that previous challenges to identify highly sparse winning tickets as subnetworks of randomly initialized dense networks (Frankle et al., 2020; Ramanujan et al., 2020) can be explained by algorithmic limitations rather than fundamental problems with LT existence. While slightly discouraging, these result on our benchmark data are coherent with reported as well as reproduced classification results on image data. This shows that our benchmarks, while artificial in nature, reflect realistic conditions that result in similar trends as real world image data sets would. Moreover, we have shown that our planting framework can also be used in a real data setting to answer a limited set of questions. For instance, by planting a trained weak ticket back into a CNN, we established that the failure of EDGE-POPUP to discover extremely sparse strong lottery tickets is likely an algorithmic rather than a fundamental limitation. This exemplifies how our framework enables experiments beyond relative method comparisons, as typically conducted on standard image benchmark data. As our results indicate, several major questions pertaining to neural network pruning are still open: How can pruning approaches for weak tickets be improved to discover tickets of best possible sparsity? How can we find weak tickets of high sparsity that match the performance of the large network without intermediate training rounds? And how can we discover highly sparse strong lottery tickets? We anticipate that our contribution can be used and extended to measure progress regarding these questions against independent sparse and well-performing baseline tickets.

## REPRODUCIBILITY

The code for our experiments is available in the Github repository RELATIONALML/PLANTNSEEK, which can be accessed with the following url: `https://github.com/RelationalML/PlantNSeek/releases/tag/v1.0-beta`. Pseudocode for the discussed planting algorithm is also available in the supplement alongside the proofs of our theoretical statements and a derivation of our planted solutions.

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

## A  THEORY

In the following section, we present the proofs of the theorems and lemmas of the main manuscript.

## A.1 Error propagation: Proof of Lemma 1

**Statement.** *Assume $\epsilon > 0$ and let the target network $f$ and its approximation $f_\epsilon$ have the same architecture. If every parameter $\theta$ of $f$ and corresponding $\theta_\epsilon$ of $f_\epsilon$ in layer $l$ fulfils $|\theta_\epsilon - \theta| \leq \epsilon_l$ for*

$$\epsilon_l := \epsilon \left( L\sqrt{m_l} \left( 1 + \sup_{x \in [-1,1]^{n_0}} \left\| x^{(l-1)} \right\|_1 \right) \prod_{k=l+1}^{L} \left( \left\| W^{(l)} \right\|_\infty + \epsilon/L \right) \right)^{-1},$$

*then it follows that $\| f - f_\epsilon \|_\infty \leq \epsilon$.*

*Proof.* Our objective is to bound $\| f - f_\epsilon \|_\infty \leq \epsilon$. We frequently use the triangle inequality and that $|\phi(x) - \phi(y)| \leq |x - y|$ is Lipschitz continuous with Lipschitz constant 1 to derive

$$\left\| x^{(l)} - x_\epsilon^{(l)} \right\|_2 \leq \left\| h^{(l)} - h_\epsilon^{(l)} \right\|_2$$
$$\leq \left\| \left( W^{(l)} - W_\epsilon^{(l)} \right) x^{(l-1)} \right\|_2 + \left\| b^{(l)} - b_\epsilon^{(l)} \right\|_2 + \left\| W_\epsilon^{(l)} \left( x^{(l-1)} - x_\epsilon^{(l-1)} \right) \right\|_2$$
$$\leq \epsilon_l \sqrt{m_l} \sup_{x \in [-1,1]^{n_0}} \left\| x^{(l-1)} \right\|_1 + \epsilon_l \sqrt{m_l} + \left( \left\| W^{(l)} \right\|_\infty + \epsilon_l \right) \left\| \left( x^{(l-1)} - x_\epsilon^{(l-1)} \right) \right\|_2$$

with $\epsilon_l \leq \epsilon/L$. $m_l$ denotes the number of parameters in layer $l$ that are smaller than $\epsilon_l$ and $\| W \|_\infty = \max_{i,j} |w_{i,j}|$. Note that $m_l \leq n_l k_{l,\max}$. The last inequality follows from the fact that all entries of the matrix $\left( W^{(l)} - W_\epsilon^{(l)} \right)$ and of the vector $(b^{(l)} - b_\epsilon^{(l)})$ are bounded by $\epsilon_l$ and maximally $m_l$ of these entries are non-zero. Furthermore, $\left\| W_\epsilon^{(l)} \right\|_\infty \leq (\left\| W^{(l)} \right\|_\infty + \epsilon_l)$ follows again from the fact that each entry of $\left( W^{(l)} - W_\epsilon^{(l)} \right)$ is bounded by $\epsilon_l$.

Thus, at the last layer it holds for all $x \in [-1,1]^{n_0}$ that

$$\| f(x) - f_\epsilon(x) \|_2 = \left\| x^{(L)} - x_\epsilon^{(L)} \right\|_2$$
$$\leq \sum_{l=1}^{L} \epsilon_l \sqrt{m_l} \left( 1 + \sup_{x \in [-1,1]^{n_0}} \left\| x^{(l-1)} \right\|_1 \right) \prod_{k=l+1}^{L} \left( \left\| W^{(l)} \right\|_\infty + \epsilon/L \right) \leq L \frac{\epsilon}{L} = \epsilon,$$

using the definition of $\epsilon_l$ in the last step. $\qquad \square$

## A.2 Existence of sparse lottery tickets: Proof of Theorem 2

Next, we prove the following lower bound on the probability that a very sparse lottery ticket exists.

**Statement.** *Assume that $\epsilon \in (0,1)$ and a target network $f$ with depth $L$ and architecture $\bar{n}$ are given. Each parameter of the larger deep neural network $f_0$ with depth $L$ and architecture $\bar{n}_0$ is initialized independently, uniformly at random with $w_{ij}^{(l)} \sim U\left( \left[ -\sigma_w^{(l)}, \sigma_w^{(l)} \right] \right)$ and $b_i^{(l)} \sim U\left( \left[ -\prod_{k=1}^{l} \sigma_w^{(k)}, \prod_{k=1}^{l} \sigma_w^{(k)} \right] \right)$. Then, $f_0$ contains a rescaled approximation $f_\epsilon$ of $f$ with probability at least*

$$\mathbb{P}\left( \exists f_\epsilon \subset f_0 : \| f - \lambda f_\epsilon \|_\infty \leq \epsilon \right) \geq \prod_{l=1}^{L} \left( 1 - \sum_{i=1}^{n_l} (1 - \epsilon_l^{k_i})^{n_{l,0}} \right),$$

*where $\epsilon_l$ is defined as in Eq. (1) and the scaling factor is given by $\lambda = \prod_{l=1}^{L} 1/\sigma_w^{(l)}$.*

*Proof.* As shown by Fischer et al. (2021), the scaling of the output by $\lambda$ simplifies the above parameter initialization to an equivalent setting, in which each parameter is distributed as $w_{ij}, b_i \sim U[-1,1]$, while the overall output is scaled by the stated scaling factor $\lambda$, again assuming that all parameters are bounded by $1 - \epsilon$. Each parameter in Layer $l$ needs to be approximated up to error $\epsilon_l$ according to Lemma 1. To match the same sparsity level of $f$, for each neuron $i$ in each Layer $l$ of $f$, we have to find exactly one neuron in the same layer (Layer $l$) of $f_0$ that represents $i$. We start

with matching neurons at Layer 1 (given the input in Layer 0) and proceed iteratively by matching neurons in Layer $l$ given the already matched neurons in Layer $l - 1$.

Let us pick a random neuron in Layer $l$ of $f_0$. How high is the probability that it is a match with a given target neuron $i$ in layer $l$ of $f$? The neuron $i$ consists of $k_i$ parameters that have to be matched. Since the corresponding neurons in layer $l - 1$ of $f$ and $f_0$ have already been matched according to our assumption, we only have one possible candidate $\theta_0$ for each of the $k_i$ parameters $\theta_i$. For uniformly distributed parameters, we have $|\theta_i - \theta_0| \leq \epsilon_l$ with probability $\epsilon_l$. For normally distributed $\theta_0 \sim \mathcal{N}(0, 1)$, the probability is at least $\epsilon_l/2$ (as long as $|\theta_0 \pm \epsilon| \leq 1$. This can be seen by Taylor approximation of the cdf of a standard normal $\Phi(z + \Delta z) - \Phi(z - \Delta z)$ in $z$. For the remainder of the proof, however, we assume uniformly distributed parameters. Thus, all $k_i$ independent parameters are a match with probability $\epsilon_l^{k_i}$. Accordingly, none of the available $n_{l,0}$ neurons in Layer $l$ is a match with probability $\left(1 - \epsilon_l^{k_i}\right)^{n_{l,0}}$.

With the help of a union bound we can deduce that the probability that at least one of the neurons $i$ in Layer $l$ of $f$ has no match in $f_0$ is smaller or equal to $\sum_{i=1}^{n_l} \left(1 - \epsilon_l^{k_i}\right)^{n_{l,0}}$. Therefore, the converse probability that we find a match for every single neuron in Layer $l$ of $f$ is at least $1 - \sum_{i=1}^{n_l} \left(1 - \epsilon_l^{k_i}\right)^{n_{l,0}}$.

Since we have to guarantee a match for each single layer and the matching probability of a new layer is conditional on the previous layer, we obtain a lower bound on the existence probability of a lottery ticket by multiplying the layerwise bounds. $\qquad\square$

This bound is only practical for very sparse target networks $f$ with neurons of small in-degrees $k_i$. It still shows that the existence of very sparse lottery tickets is possible under the right conditions. To provide an intuition how this bound on the existence probability depends on the relevant parameters, we visualize the bound when one component is varied in a simple example, in which all nodes and layers are homogeneous so that they have identical properties like degree, width, etc. Fig. 6 shows that the bound most critically depends on the degree of a node and the width of the mother network. Yet, all parameters matter and can make the existence of a LT unlikely. Extreme sparsity can pose significant challenges to pruning algorithms, as we also see in our experiments with planted solutions. Inspired by this proof, we therefore explain next how to plant tickets (which could have variable sparsity levels).

### A.3 PLANTING ALGORITHM

The idea of layerwise matching neurons starting with the layer closest to the input can be transferred to planting. The approach applies to general initial networks $f_0$ whose parameters have not necessarily been randomly initialized and captures the full variability of possible target solutions by allowing for different scaling factors per neuron. Why are scaling factors necessary? The target network and the initial mother network need to produce output that lies in a similar range (ideally the one of the training labels). Yet, the target network has to achieve this by adding up a much smaller number of parameters spanning a larger range. A different perspective on the same issue is that a pruned lottery ticket needs to be scaled up to compensate for the lost parameters. Note the scaling factor $\lambda$ in Theorem 2 for that purpose. To cover natural degrees of freedom in ReLU networks, we instead allow for different neuron-wise scaling factors $\lambda_j > 0$. Note that such a scaling can be compensated by rescaling of parameters in the next layer, as

$$x_k^{(l+1)} = \phi \left[ \sum_i (w_{kj}^{(l+1)}/\lambda_j) \phi \left( \sum_i (w_{ji}^{(l)} \lambda_j) x_i^{(l-1)} + (b_j^{(l)} \lambda_j) \right) + b_k^{(l+1)} \right].$$

To apply only a small change to the random mother network during planting, we choose $\lambda_j$ together with matching target neurons and mother network neurons. We search in each layer of $f_0$ for suitable neurons that best match a target neuron in $f$, starting from the first hidden layer. Given that we matched all neurons in layer $l - 1$, we try to establish their connections to neurons in the current layer $l$. A best match is decided by minimizing the l2-distance to its potential input parameters thereby adjusting for an optimal scaling factor. For example, let neuron $i$ in Layer $l$ of the target $f$ have non-zero parameters $(b, \boldsymbol{w})$ that point to already matched neurons in Layer $l - 1$. With each of

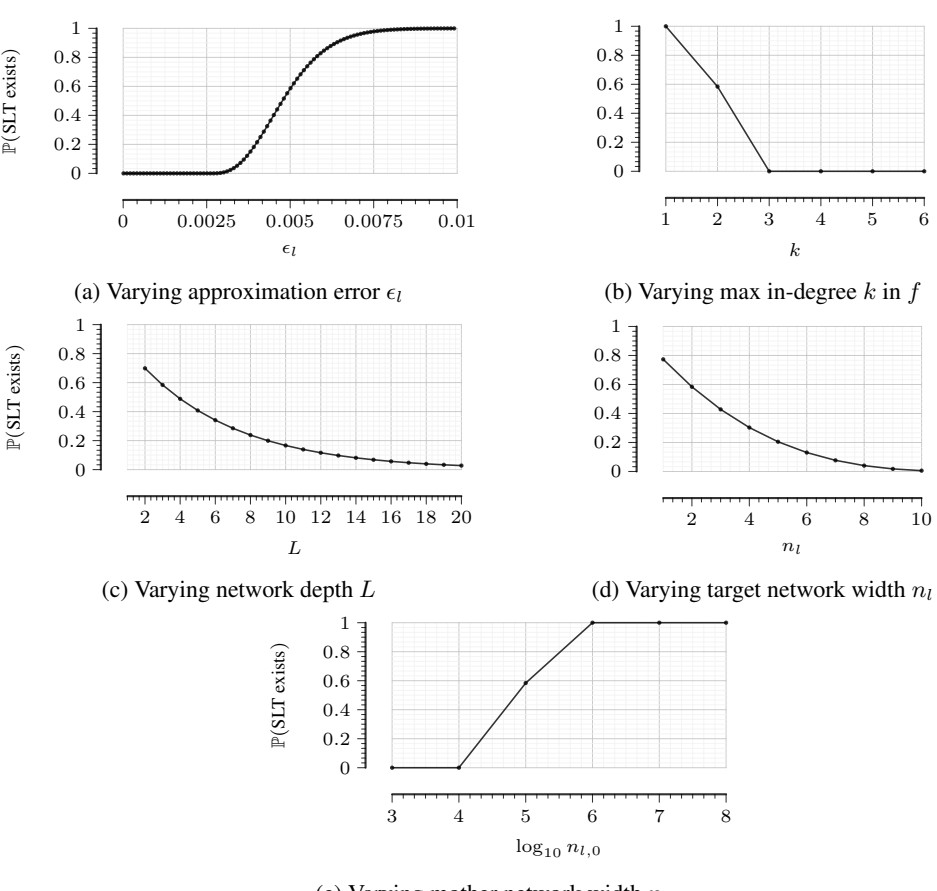

(a) Varying approximation error $\epsilon_l$

(b) Varying max in-degree $k$ in $f$

(c) Varying network depth $L$

(d) Varying target network width $n_l$

(e) Varying mother network width $n_{l,0}$

Figure 6: *Visualization of lower bound.* Bound on SLT existence probability Eq. (A.2) for $\epsilon_l = 0.005$, $k_i = 2$, $L = 3$, $n_l = 2$, and $n_{l,0} = 10^5$. In each plot, one single variable is varied while the remaining ones are kept fixed.

---

**Algorithm 1:** Planting

---

**input** : target $f$, larger neural network $f_0$
**output:** $f_0$ with planted $f$ ($f \subset f_0$), output scaling factors $\boldsymbol{\lambda}$
1  Initialize $\boldsymbol{\lambda}_{\text{old}} = [1]^{n_0}$                                    // scaling factors for input are 1
2  **for** $l = 1$ *to* $L - 1$ **do**
3       **for** *all neurons $i$ of $f$ in Layer $l$* **do**
4           $\boldsymbol{\theta} := (b, \boldsymbol{w}\boldsymbol{\lambda}_{\text{old}})$                         // scaled parameters of neuron $i$ in $f$
5           $\boldsymbol{m}^* = \arg\min_{\boldsymbol{m}} q_{\theta}(\boldsymbol{m})$                    // find best match for $i$ in $f_0$
6           Replace $\boldsymbol{m}^*$ in $f_0$ by $\boldsymbol{\theta}/\lambda(\boldsymbol{m}^*)$
7           $\lambda_i = \lambda(\boldsymbol{m}^*)$                          // remember scaling factor of $i$ in $f_0$
8       **end**
9       $\boldsymbol{\lambda}_{\text{old}} = \boldsymbol{\lambda}$
10 **end**
11 **return** $f_0, \boldsymbol{\lambda}$

---

**Algorithm 2:** Faster planting by random matching

---

**input** : target $f$, larger neural network $f_0$
**output:** $f_0$ with planted $f$ ($f \subset f_0$), output scaling factors $\boldsymbol{\lambda}$
1  Initialize $\boldsymbol{\lambda}_{\text{old}} = [1]^{n_0}$                                    // scaling factors for input are 1
2  **for** $l = 1$ *to* $L - 1$ **do**
3       **for** *all neurons $i$ of $f$ in Layer $l$* **do**
4           $\boldsymbol{\theta} := (b, \boldsymbol{w}\boldsymbol{\lambda}_{\text{old}})$                               // scaled parameters of $i$ in $f$
5           $\boldsymbol{m}^* =$ parameters of random unmatched neuron in $f_0$
6           Replace $\boldsymbol{m}^*$ in $f_0$ by $\boldsymbol{\theta}/\lambda(\boldsymbol{m}^*)$
7           $\lambda_i = \lambda(\boldsymbol{m}^*)$                          // remember scaling factor of $i$ in $f_0$
8       **end**
9       $\boldsymbol{\lambda}_{\text{old}} = \boldsymbol{\lambda}$;
10 **end**
11 **return** $f_0, \boldsymbol{\lambda}$

---

the matched neurons $j'$ in Layer $l - 1$ has been previously associated a scaling factor $\lambda_{\text{old},j'}$ so that the corrected $\boldsymbol{\theta} = (b, \boldsymbol{w}\boldsymbol{\lambda}_{\text{old}})$ parameters would compute the correct neuron in $f_0$. In Layer $l$ of $f_0$, each neuron $j$ that we have not matched yet could be a potential match for $i$. Let the corresponding parameters of $j$ be $\boldsymbol{m}$. The match quality between $i$ and $j$ is assessed by

$$q_{\theta}(m) = \|\boldsymbol{\theta} - \lambda(m)\boldsymbol{m}\|_2 \,,$$

where $\lambda(m) = \boldsymbol{\theta}^T \boldsymbol{m}/\|\boldsymbol{m}\|_2^2$ is the optimal scaling factor. The best matching parameters $m^* = \arg\min_m q_{\theta}(m)$ are replaced by rescaled target parameters $\theta/\lambda(m^*)$ in $f_0$ and we remember the scaling factor $\lambda(m^*)$ to consider matches of neurons in Layer $l + 1$. Note that this rescaling is necessary to ensure that the neuron is properly hidden and attains similar values as other non-planted neurons in $f_0$. In addition to the provided pseudocode (Algorithm 1), a Python implementation is provided online.[1]

Matching neurons thoroughly can be computational resource intensive, if the target network $f$ consists of a high number of neurons, because each neuron needs to be compared with most of the neurons in the mother network or at least a significant share of candidate neurons. A fast alternative is to pick a random neuron in the mother network as a match and choose an appropriate scaling factor (see Algorithm 2).

### A.4 CONSTRUCTION OF TARGETS FOR PLANTING

We propose three toy examples for planting lottery tickets that pose different challenges for pruning algorithms. Here, we explain the main ideas behind their construction in more detail. A Python

---

[1]`www.github.com/RelationalML/PlantNSeek`

implementation of related regression and classification problems is provided alongside the supplementary manuscript.

### A.4.1 RELU UNIT

Apart from a trivial function $f(x) = 0$, a univariate ReLU unit $f(x) = \phi(x) = \max(x, 0)$ is the most sparse lottery ticket that is possible. Assuming a mother network $f_0$ of depth $L$, a ReLU can be implemented with a single neuron per layer. Any path through the network with positive weights $\prod_{l=1}^{L} \phi(w_{i_{l-1}i_l}x)$ defines a ReLU with scaling factor $\lambda = \prod_{l=1}^{L} w_{i_{l-1}i_l}$ for indices $i_l$ in Layer $l$ with $w_{i_{l-1}i_l} > 0$.

Note that each random path fulfills this criterion with probability $0.5^L$ so that even random pruning could have a considerable chance to find an optimal ticket. A winning path exists with probability $\prod_{l=1}^{L}(1 - 0.5^{n_{l,0}})$, which is almost 1 even in small mother networks. Thus, planting is not really necessary in this case. Since not all pruning algorithms set biases to zero, however, we still set all randomly initialized biases along a winning path to zero to make the problem easier.

As we see in experiments, despite this simplification, pruning algorithms are severely challenged in finding an optimally sparse ticket. Even though basic, a ReLU unit seems to be a suitable benchmark that is a common building block of other tickets.

### A.4.2 CIRCLE

For simplicity, we restrict ourselves to a 4-class classification problem with 2-dimensional input. The output is therefore 4-dimensional, where each output unit $f_c(x)$ corresponds to the probability $f_c(x)$ that an input $(x_1, x_2) \in [-1, 1]^2$ belongs to the corresponding class $c$ with $c = 0, 1, 2, 3$. As common, this probability is computed assuming softmax activation functions in the last layer. The decision boundaries are defined in the last layer based on inputs of the form $g(x_1, x_2)$. The role of the first layers with ReLU activation functions of a Circle target $f$ is to compute the function $g(x_1, x_2) = x_1^2 + x_2^2$, which is fundamental to many problems, in particular to the computation of radial symmetric functions.

The high symmetry of $g(x_1, x_2)$ allows us to construct a particularly sparse representation by mirroring data points along axes as visualized in Figure 2 (a). With the first two layers ($l = 1, 2$), we map each input vector $(x_1, x_2)$ to the first quadrant by defining $x_1^{(1)} = \phi(x_1) + \phi(-x_1)$ and $x_2^{(1)} = \phi(x_2) + \phi(-x_2)$. Thus, Layer $l = 1$ consists of 4 neurons, i.e., $x_1^{(1)} = \phi(x_1)$, $x_2^{(1)} = \phi(-x_1)$, $x_3^{(1)} = \phi(x_2)$, $x_4^{(1)} = \phi(-x_2)$, while Layer $l = 2$ consists of 2 neurons, i.e., $x_1^{(2)} = \phi\left(x_1^{(1)} + x_2^{(1)}\right)$, $x_2^{(2)} = \phi\left(x_3^{(1)} + x_4^{(1)}\right)$.

Each consecutive layer $l$ mirrors the previous layer $(x_1^{(l-1)}, x_2^{(l-1)})$ along the axis $\boldsymbol{a}^{(l)} = (\cos(\pi/2^{l-1}), \sin(\pi/2^{l-1}))$. It achieves this by mapping the neurons of the previous layer to three neurons, one representing the component of $\boldsymbol{x}^{(l-1)}$ that is parallel to $\boldsymbol{a}^{(l)}$, and two neurons that each represent the positive or negative signal component that is perpendicular to the axis $\boldsymbol{a}^{(l)}$. The last two neurons could be added to a single neuron in the next layer if we want to decrease the width of some layers to 2 in between. To take more advantage of the allowed depth $L$, we map three neurons immediately to the next three neurons that represent the mirroring by defining

$$x_1^{(l)} = \phi\left(a_1^{(l)} x_1^{(l-1)} + a_2^{(l)} x_2^{(l-1)} - a_2^{(l)} x_3^{(l-1)}\right), \qquad x_2^{(l)} = \phi\left(a_2^{(l)} x_1^{(l-1)} - a_1^{(l)} x_2^{(l-1)} + a_1^{(l)} x_3^{(l-1)}\right),$$

$$x_3^{(l)} = \phi(-h_2^{(l)}).$$

If the depth of our network is high enough, we could use the parallel component $x_1^{(l)}$ as estimate of the radius of the input. To enable higher precision for networks of smaller depth, however, we also apply to each remaining component a piecewise linear approximation of the univariate function $h(x) = x^2$ and add those two components. Note that any univariate function can be easily approximated by a neural network of depth $L = 2$. The precise approach is explained in our next example.

### A.4.3 HELIX

To test the ability of pruning algorithms to detect lower dimensional submanifolds, we approximate a helix with three output coordinates $f_1(x) = (5\pi + 3\pi x) * \cos(5\pi + 3\pi x)/(8\pi)$, $f_2(x) = (5\pi + 3\pi x) * \sin(5\pi + 3\pi x)/(8\pi)$, and $f_3(x) = (5\pi + 3\pi x)/(8\pi)$ for 1-dimensional input $x \in [-1, 1]$. As we have observed that many pruning algorithms have the tendency to keep a higher number of neurons closer to the input (and sometimes also the output layer), we construct a ticket that has similar properties. This should ease the task for pruning algorithms to find the planted winning ticket.

Each of the components $f_i(x)$ is an univariate function that we can approximate by an univariate deep neural network $n_i(x)$ that encodes a piece-wise linear function (see Figure 2 (c) for an explanation). As neural networks are generally overparameterized, we have multiple options to represent $n_i(x)$. For simplicity, we write it as composition of the identity with a depth $L = 2$ univariate network $g_i(x)$ of width $N$ in the hidden layer, which can be written as

$$g_i(x) = \sum_{j=1}^{N} a_j^{(i)} \phi(p_j(x - s_j)) + b^{(i)},$$

where the signs $p_j \in \{-1, 1\}$ can be chosen arbitrarily (and we chose alternating signs to create diversity). The knots $s = (s_j)_{j \in [N]}$ mark the boundaries of the linear regions and $a = (a_j^{(i)})_{j \in [N]}$ indicate changes in slopes $m_j^{(i)} = (f_i(s_{j+1}) - f(s_j)) / (s_{j+1} - s_j)$ (with $s_{N+1} := s_N + \epsilon$) from one linear region to the next. $a_j^{(i)} = m_j^{(i)} - m_{j-1}^{(i)}$ for $2 \leq i \leq N$, $a_1^{(i)} = m_1^{(i)}$, and $b^{(i)} = f_i(s_1) - \sum_{j=1}^{N} a_j^{(i)} \phi(p_j(s_1 - s_j))$. Note that only the outer parameters $a_j^{(i)}$ are function specific, while the inner parameters $p_j$ and $s_j$ can be shared among the functions $f_i$.

We thus create a helix ticket by first mapping the input $x \in [-1, 1]$ to $[0, 2]$. This allows us to represent the identity in the later layers by $\phi(x) = x$, as $x \geq 0$. We can always compensate for the bias $+1$ by subtracting a bias $-1$ when needed. $f_3(x) = (5\pi + 3\pi x)/(8\pi)$ can therefore be represented by a path from the input to the output that only contains a single neuron per layer. We concatenate this path with a neural network that consists of layers that approximate $f_1(x)$ and $f_2(x)$ and otherwise identity functions. At Layer $l = 2$, this network creates neurons of the form $\phi(p_j(x - s_j))$, where the knots $s_j$ mark an equidistant grid of $[0, 2]$. Layer $l = 3$ creates two neurons, one corresponding to $x_1^{(2)} = f_1(x)$ and one corresponding to $x_2^{(2)} = f_2(x)$. These can be computed by linear combination of the previous neurons using the parameters $a_j^{(i)}$ and $b^{(i)}$. All the remaining layers basically encode the identity.

## B  EXPERIMENTS

In this section, we discuss all relevant parameters and set-ups to reproduce the experimental results. Furthermore, we provide additional results obtained for singleshot learning spanning different architectures and pruning schemes. All source code to run pruning algorithms and to generate the data is made publicly available.

### B.1  HYPERPARAMETERS AND DATA

For each experiment, we generate $n = 10000$ samples, where input data is sampled from $[-1, 1]$ for ReLU and Helix and from $[-1, 1]^3$ for Circle. The output for ReLU is computed by $f(x) = max(0, x)$. For Helix, we compute the three output coordinates as $f_1(x) = (5\pi + 3\pi x) * \cos(5\pi + 3\pi x)/(8\pi)$, $f_2(x) = (5\pi + 3\pi x) * \sin(5\pi + 3\pi x)/(8\pi)$, and $f_3(x) = (5\pi + 3\pi x)/(8\pi)$. For Circle, we consider circles centered at the origin with radius $\sqrt{0.2}$, $\sqrt{0.5}$, and $\sqrt{0.7}$ as decision boundaries for the classes. We additionally introduce a small amount of noise to simulate real world data more closely. For Circle we flip approximately 1% of samples to the next closest class, and for the two regression problems we introduce additive noise drawn from $\mathcal{N}(0, 0.01)$ to each output dimension. To assess the accuracy respectively mean squared error of the tickets and trained models, we split off 10% of the data that acts as a hold out test set.

In general, all initial networks for each specific task are generated using the algorithm explained in App. B.1. For `Circle`, we use 10 knots for the piecewise linear approximation, and 30 knots for the piecewise linear approximations done in `Helix`. To prune by GRASP, SNIP, SYNFLOW, MAGNITUDE, and RANDOM and train the derived tickets, we use `Adam` Kingma & Ba (2015) with a learning rate of 0.001. We found that this learning rate performed well over all experiments, and leading to accurate models when there is no pruning. It also corresponds to the default settings suggested by the authors of SYNFLOW Tanaka et al. (2020). Training of the discovered tickets was done for 10 epochs across all experiments, where we could always observe a convergence of the respective score on the validation sets (accuracy or MSE). We measured loss by MSE respectively cross entropy loss and used a batch size of 32 for all experiments. We report obtained intervals as minimum and maximum as well as mean across 10 repetitions for multishot, and across 25 repetions for the main singleshot experiments, all measured on the hold out test set.

**Singleshot pruning** For singleshot pruning, we considered networks of depth $3, 5, 10$ each with layer width 100 for all three data sets on target sparsities $\{0.01, 0.1, 0.5, 1\}$ and the sparsity of the ground truth ticket. Additionally, we tested for a network of depth 6 and width 1000 on `Circle` for the same sparsity levels. As suggested by Tanaka et al. (2020), we also test SYNFLOW in combination with 100 rounds of pruning for a network of depth 6 and width 100 on `Circle`. For all additional singleshot experiments, we provide results in the next section.

**Multishot pruning** For multishot pruning, we alternated pruning and training for 10 rounds, where each training step was carried out for 5 epochs, which consistently lead to convergence of accuracy on the considered `Circle` data set. Similar to singleshot pruning, we considered target sparsities $\{0.01, 0.1, 0.5, 1\}$ and ground truth ticket sparsity.

**EDGE-POPUP pruning** To prune with EDGE-POPUP, we here used the parameters suggested in the original code of Ramanujan et al. (2020), which is SGD with momentum of 0.9 and weight decay 0.0005, combined with cosine annealing of the learning rate. To establish a comparison to the multishot pruning results, we train the scores for 10 epochs. Additionally, for the experiment extending EDGE-POPUP by annealing the sparsity level, we slowly reduce the sparsity over time to $\rho^{i/10}$, where $\rho$ is the desired network sparsity, and $i$ is the current epoch.

## B.2 DETAILED DISCUSSION OF RESULTS

**Singleshot pruning** In singleshot pruning, which is originally applied in SNIP, and GRASP, edges are scored in a single pass and then pruned to the desired sparsity. Compared to multishot pruning, this saves significant amounts of resources by preventing training entirely, if a strong ticket is found, or only training a small subnetwork once, in case a weak ticket is found. For our benchmark data, we construct networks of depth 5 and width 100 and test the ability of algorithms to discover both strong and weak tickets. The key results are visualized in Fig. 3, reporting performance of the algorithms trying to discover tickets at varying sparsity levels across 25 repetitions.

We find that all approaches, including RANDOM pruning, are able to find weak tickets for moderate sparsity levels for `Circle` and ReLU, but fail to recover them on the manifold learning task `Helix` entirely. Although MAGNITUDE pruning was originally not designed for this pruning strategy, it is on par with state-of-the-art singleshot methods. For lower sparsity levels $\leq 0.01$, in particular baseline ticket sparsity, all methods fail to recover good subnetworks. Dissecting the results, we observe layer collapse, meaning that entire layers are masked, thus disrupting flow through the network. Despite that SYNFLOW was proposed as a solution to this issue, we observe that it also experiences layer collapse for extreme sparsities, even when pruning for 100 rounds as suggested in the original paper, performing only slightly better than with one round of pruning (see Supp. B.5). In summary, with only a single pruning round, most pruning algorithms discover weak tickets at moderate sparsity, however fail to recover weak tickets of low sparsity and any strong tickets.

**Robustness to noise** To model real-world settings, all our datasets contain small amounts of noise as described in Supp. B. To rule out that noise in the data is the primary source for issues with discovering tickets, we generated `Circle` datasets with varying levels of noise. The results indicate that on the one hand, without noise we do not see much of an improvement in terms of discovered

tickets, but on the other hand observe that the algorithms are robust to even large amounts of noise, finding tickets with almost similar performance as with no noise at all (see Supp. B.5).

**Comparison to results on image data** The reported results are in line with experiments on image data as reported in the literature (Tanaka et al., 2020). In particular, for all these methods a similar drop around 0.01 sparsity is observed for different VGG and Resnet architectures and image data sets. Similarly, layer collapse has been reported for image data. The main difference is that for our data, we know the obtainable sparsity as well as performance of tickets, setting these results into a context beyond trendline differences of methods for selected sparsity values.

### B.3   PRUNING COMBINED WITH TRAINING

While much more resource intensive, iteratively training followed by pruning and resetting to initial weights, slowly annealing to the desired sparsity, has been proven a successful approach to discover lottery tickets. Here we extend this pruning scheme to other approaches beyond magnitude pruning.

**Multishot pruning** To investigate the effect of multishot pruning, we run each of the previous methods iteratively for 10 rounds on our benchmark datasets (see Fig. 4). Analogous to iterative magnitude pruning, for each round $r$, we iteratively reduce the sparsity to $\rho^{r/10}$, where $\rho$ is the desired network sparsity. Within each round, the current subnetwork is first shortly trained, then pruned to the current target sparsity, and then reset to initial parameters for the next round. Compared to the singleshot results, we observe that for the classification task, MAGNITUDE, SNIP, and SYNFLOW are able to retrieve weak tickets of much higher sparsity. Furthermore, these three approaches are now able to recover weak tickets of moderate sparsities also for the challenging `Helix` dataset. Overall, SYNFLOW consistently performs best in discovering weak tickets, even recovering the extremely sparse baseline ticket for `Circle`. We also observe that GRASP performs poor overall, noting that it is a method designed for singleshot pruning. Examining the results, we see that GRASP experiences layer collapse already in early iterations with large target sparsities. None of the above approaches is able to discover a strong ticket.

To discover strong tickets, Ramanujan et al. (2020) proposed EDGE-POPUP, which falls in the same category of multishot pruning approaches. EDGE-POPUP assigns each model parameter a score value, which is then actively trained for several rounds while freezing all original parameters, requiring a similar computational effort as multishot pruning. Training EDGE-POPUP for 10 rounds, we observe that it discovers a strong ticket of sparsity 0.5 for `Circle` however fails to discover tickets of different sparsity, which is in line with their original results Ramanujan et al. (2020). Similar to other algorithms, we observe layer collapse. We can extend their original algorithm using annealing as in multishot pruning by slowly decreasing the sparsity in every round, which increases performance allowing it to discover a subnetwork with reasonable accuracy at 0.1 sparsity. Interestingly, EDGE-POPUP is not able to find any good subnetwork for the regression tasks.

**GRASP with local sparsity constraints** As observed before, GRASP seems unsuitable for multishot pruning due to early layer collapse. Several works (You et al., 2020; Tanaka et al., 2020) considered local sparsity constraints, having a target sparsity for each layer, or even channel. These however impose unrealistic architecture constraints as layer sparsity is usually imbalanced (Tanaka et al., 2020), which also holds true for our `Circle` and `Helix` benchmarks. With the goal to avoid layer collapse, we still equipped GRASP with local sparsity constraints per layer (see Supp. B.5). Yet, the flow through the layers stays interrupted. A possible explanation is that GRASP incorporates information about weight couplings via the hessian in its pruning strategy, which makes it more sensitive to removing individual connections from the mask as it happens during iterative pruning.

**Comparison to results on image data** Our results are coherent with the reported relative trends on image tasks both for strong and weak tickets (Ramanujan et al., 2020; Tanaka et al., 2020). We further reproduced results for VGG16 on CIFAR10 with non-zero bias initialization (Fig. 5 left). In particular, for strong tickets we observe the same trends for EDGE-POPUP spiking at 0.5 sparsity, performing less well at sparsity 0.1, and not recovering tickets for higher and lower sparsity. Again consistent with previously reported results, other methods are neither suited nor designed to find strong tickets. For weak tickets, SYNFLOW performs best, with only a slight margin towards SNIP and MAGNITUDE. This margin is however much tighter then originally reported in SYNFLOW, as we allow both SNIP and MAGNITUDE to learn in a multishot fashion, which closely resembles the approach of iterative magnitude pruning, so that we do not observe layer collapse for these methods.

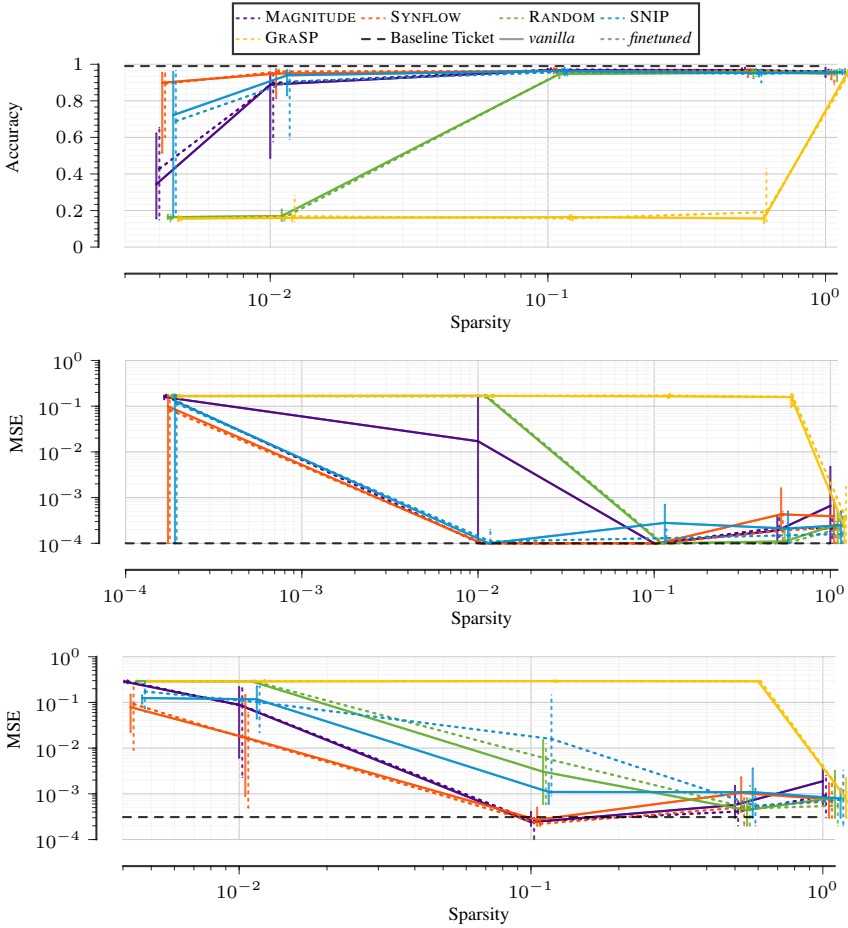

Figure 7: *Finetuning vs initialization.* Performance for `Circle` (top), `ReLU` (middle), and `Helix` (bottom) for 10 rounds of alternating pruning and training. Visualized are weak ticket performances (i.e. training on initialized weights) against finetuned subnetworks (i.e. no reset of weights after final pruning round). Baseline ticket performance in black.

**VGG with strong tickets** Finally, to put the hypothesis to a test that EDGE-POPUP is limited to discover strong tickets of suboptimal sparsity of around 0.5, we investigate its capabilities to recover a planted baseline ticket from VGG16. For that, we use SYNFLOW to discover a weak ticket of sparsity 0.01 from VGG16 with multishot pruning, trained the weak ticket, and planted it back into the network. Running EDGE-POPUP on this network, we observe that it indeed cannot retrieve the baseline ticket of desired sparsity in this real world setting (see Fig. 5 right).

## B.4 OTHER PRUNING APPROACHES

**Finetuning** Recent results indicated that finetuning discovered subnetworks yields better models than training these subnetworks from scratch (Liu et al., 2021a). In particular, they proposed to use iterative (magnitude) pruning and skip the resetting of the parameters to their initial values, but rather continue training the current parameter set, which incurs no additional computational overhead. Here, we investigate how finetuning affects results on our benchmark data for all considered methods. We repeat the multishot learning experiments in combination with finetuning and visualize the results in Fig. 7. For the considered classification task, finetuning matches the performance of classical 'weak ticket'-training with parameter reset, achieving almost perfect accuracy levels for up to .01 target sparsity. For even sparser target networks, we see a marginal improvement of MAG-NITUDE and SYNFLOW by using finetuning, which is however within the confidence of the original prediction. For the regression tasks we get a slightly different view: here, the original weak tick-

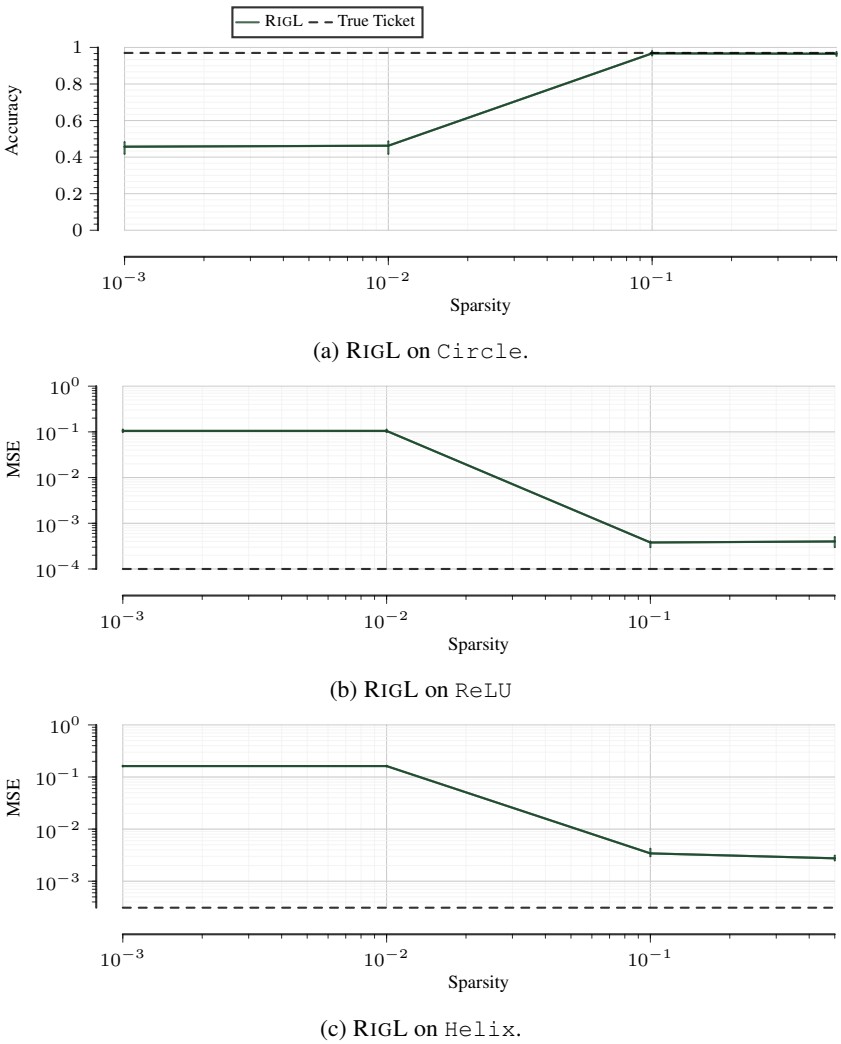

(a) RIGL on `Circle`.

(b) RIGL on `ReLU`

(c) RIGL on `Helix`.

Figure 8: RIGL. Performance for `Circle` (top), `ReLU` (middle), and `Helix` (bottom) for 60 rounds of training with default parameters. We report mean and minimum and maximum values across 10 repetitions. Baseline ticket performance is indicated as black dashed line.

ets already detoriate in performance for relatively dense target networks of .5 or .1 target sparsity. Notably, for the `ReLU` task, finetuning does improve ticket performance for those sparsity levels, such that the algorithms can match the performance of the ground truth ticket. For more extreme sparsity levels, we see a drastic improvement for MAGNITUDE pruning, but no improvement for other algorithms or at ground truth ticket performance. For `Helix`, the results are mixed, where we observe an improvement at dense target sparsities of .5 similar to before, but no improvement or even a decrease of performance for smaller target sparsity levels. In summary, on our benchmarks, finetuning helps to improve performance at relatively large target sparsity levels ($> .1$), but does not provide an advantage compared to the common training after parameter reset for extreme sparsities.

**RIGL** While the main focus of our paper are lottery tickets, we here briefly discuss results for RIGL (Evci et al., 2020), a state-of-the-art dynamic sparse training approach, which results in sparsified and trained network architectures which are comparable to trained 'weak' tickets. For our benchmark data, we construct similar networks as for the multishot experiments – i.e. depth 6 and width 100 fully connected networks – and run the available implementation of RIGL with default parameters as suggested in the paper, and Adam optimization with the same parameter settings as for all other considered methods. We train networks for 60 epochs, which results in a comparable amount

of training time as in the multishot experiments, and note that good performance is reached much earlier.

The original results reported in Evci et al. (2020) indicate that for their considered (classification) tasks, RIGL outperforms other dynamic sparse training methods, and that for target sparsity levels of .1 and lower, performance quickly deteriorates for all methods.[2] In the original paper, there was no exploration of the more extreme sparsities considered here, nor a comparison to ticket pruning other than SNIP. Our results on `Circle` match those results, with RIGL being able to match ground truth ticket performance for sparsity .5 and .1. The performance, however, decreases quickly with more extreme sparsity levels (see Fig. 8). In comparison with the multishot results, we observe that SYNFLOW, SNIP, and MAGNITUDE pruning outperform RIGL on this task for the extreme sparsity levels (compare Fig. 4). Note that the version of SNIP used in the original paper is essentially the singleshot approach, which indeed performs worse than RIGL (compare Fig. 3). For regression tasks, we see a similar trend, with RIGL performing comparably good as SNIP for sparsity levels $\geq .1$, but the performance decreases rapidly for more extreme target sparsitiy levels. Generally, for the regression tasks we observe that RIGL is outperformed by the state-of-the-art ticket pruner SYNFLOW and iterative MAGNITUDE pruning. Yet, RIGL allows for efficient computations, saving FLOPS by only infrequently updating gradients, which render it the method of choice for target sparsities of $\geq .1$ in specific applications.

### B.5 ADDITIONAL RESULTS ON SINGLESHOT LEARNING

**Model depth**    In this section, we provide all results from the singleshot learning for depths $3, 5, 10$ and width $100$ in Figure 9,10, and 11, respectively. We observe that all methods have problems to deal with smaller networks, while the results for the larger networks are consistent.

**Model width**    To investigate the effect of layer size, we run an experiment with a network of depth 6 and width 1000 on `Circle` – as the layer size is an important factor for the theoretical probability of the existence of tickets – and report the results in Figure 12. We observe that although the network is much larger, there is barely any change in comparison to the previous results. Note that the results after training of tickets of individual sparsities cannot be directly compared directly to the other singleshot results, as the tickets are much larger (due to the much larger number of parameters) and hence easier to train.

**Multiple pruning rounds**    We report the results of running SYNFLOW with 100 rounds of pruning on a network of depth 6 and width 100 for `Circle` in Figure 13. We find that there is again barely any change to the original singleshot results after pruning, but there is a slight increase in performance after training compared to the single-round singleshot results.

**Noise experiments**    To test the robustness of pruning algorithms to noise in the data, we considered the `Circle` problem with a network of depth 6 and width 100 and varied the amount of noise in the data to be $\{0, 0.001, 0.01, 0.1\}$. We report the results before and after training in Fig. 14.

### B.6 ADDITIONAL RESULTS ON MULTISHOT LEARNING

To test whether we can reach an improvement of the performance of tickets discovered by GRASP using a multishot pruning approach and avoid layer collapse, we ran additional experiments using a local pruning rate. In particular, we used layer-wise pruning setting the target sparsity of the parameters of each individual layer to the global sparsity level. We report results in Fig. 15.

---

[2]Note that Evci et al. (2020) use percentage of pruned parameters for their plots, i.e. $1-$sparsity compared to our paper.

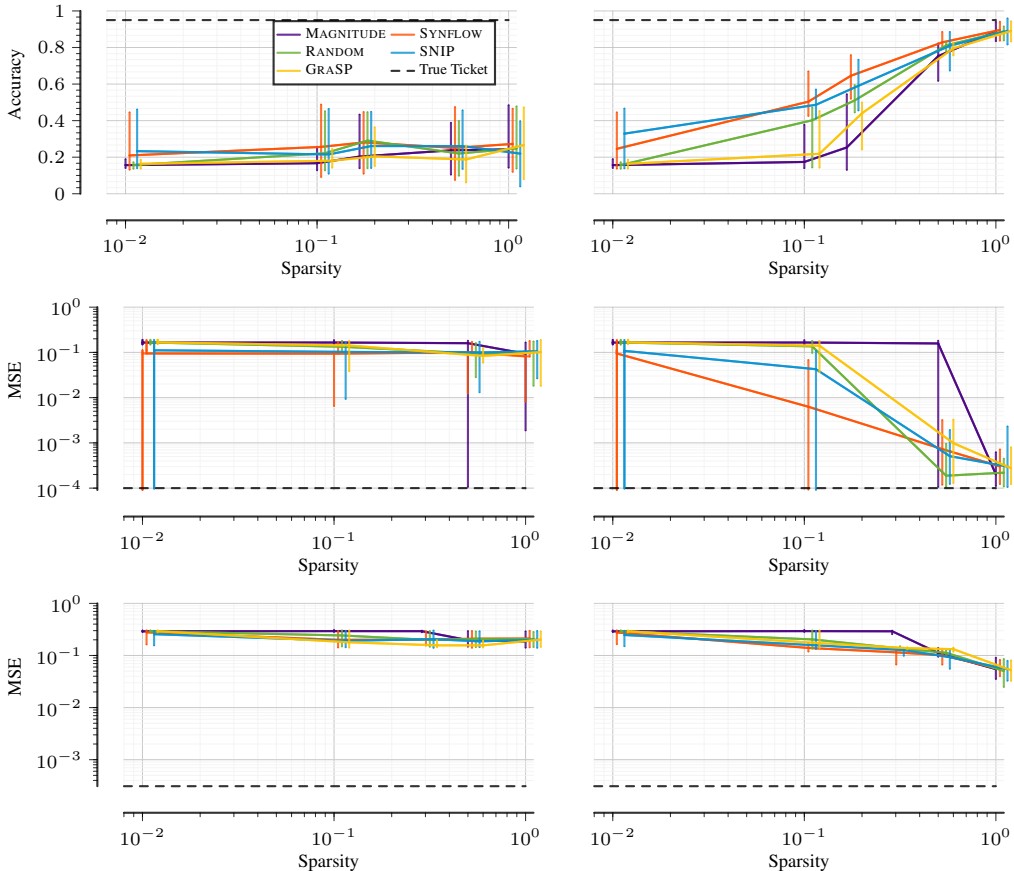

Figure 9: *Singleshot results depth* 3. Performance of discovered tickets for `Circle`, `ReLU`, and `Helix` against target sparsities as mean and obtained intervals (minimum and maximum) across 25 runs. In order of appearance from top to bottom: `Circle`, `ReLU`, and `Helix` post pruning (left) and post training performance (right). Baseline tickets have sparsities of .16, .01, and .29, and their performance is given by black dashed line.

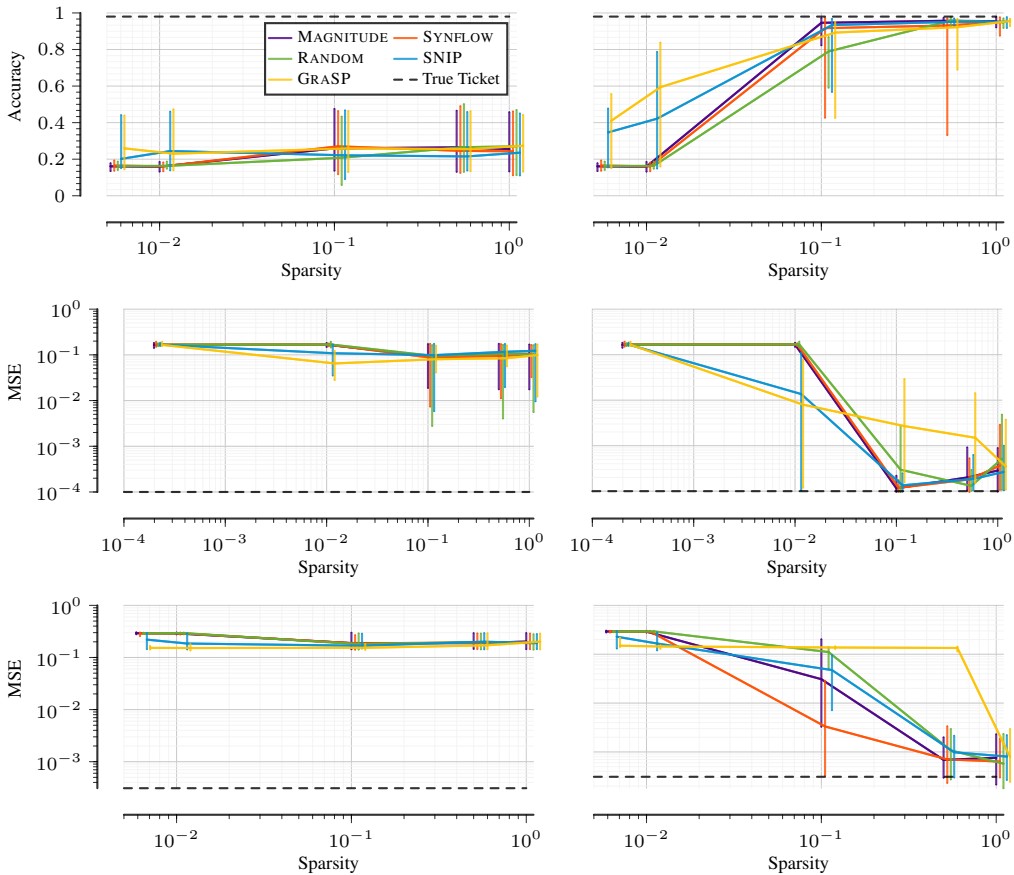

Figure 10: *Singleshot results depth* 5. Performance of discovered tickets for `Circle`, `ReLU`, and `Helix` against target sparsities as mean and obtained intervals (minimum and maximum) across 25 runs. In order of appearance from top to bottom: `Circle`, `ReLU`, and `Helix` post pruning (left) and post training performance (right). Baseline ticket with leftmost sparsity and performance given by black dashed line.

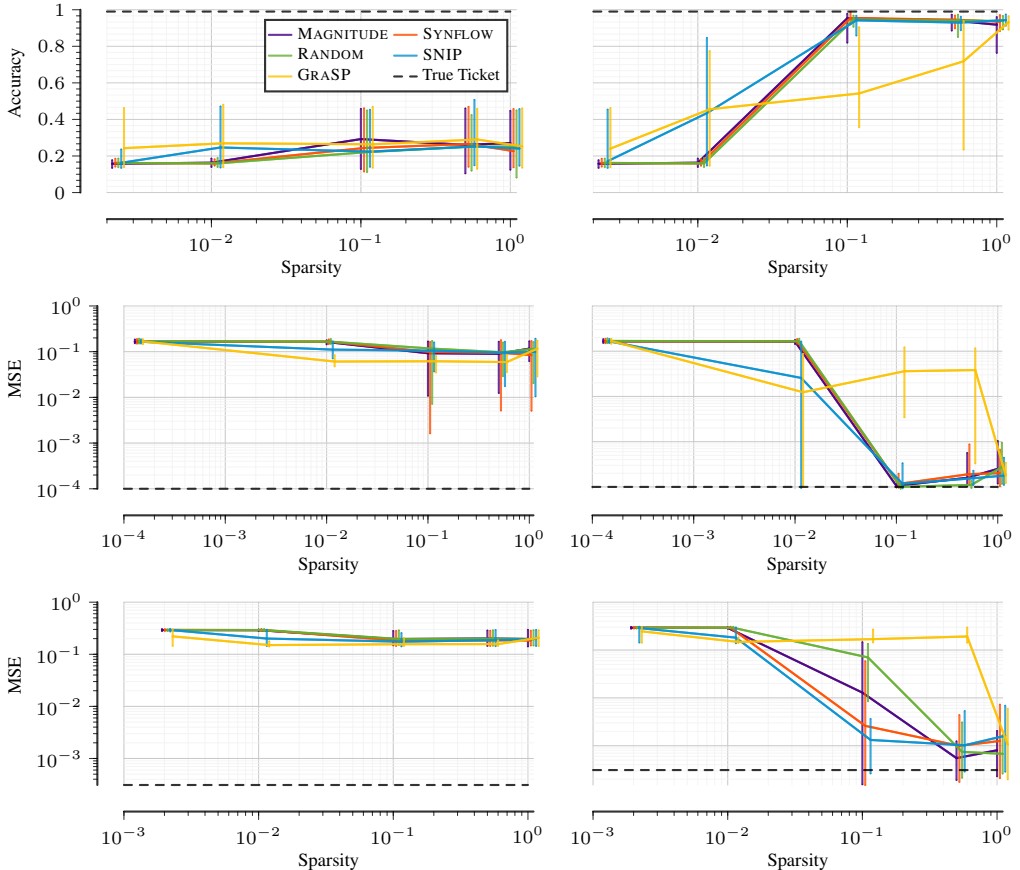

Figure 11: *Singleshot results depth* 10. Performance of discovered tickets for `Circle`, `ReLU`, and `Helix` against target sparsities as mean and obtained intervals (minimum and maximum) across 25 runs. In order of appearance from top to bottom: `Circle`, `ReLU`, and `Helix` post pruning (left) and post training performance (right). Baseline ticket has leftmost sparsity and its performance given by black dashed line.

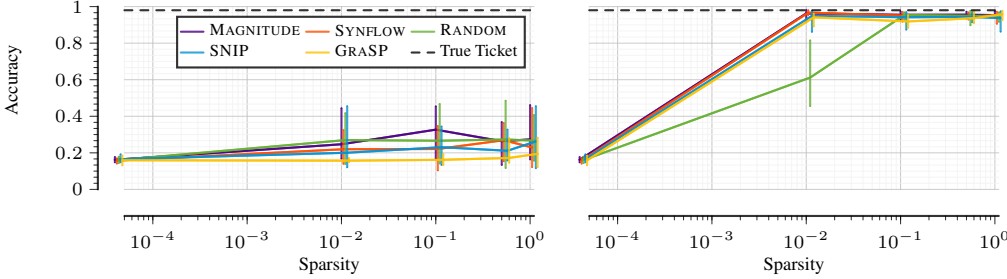

Figure 12: *Singleshot results, depth* 6 *width* 1000. Performance on test data are plotted for `Circle` against target sparsities. We report mean and obtained intervals (minimum and maximum) across 10 repetitions of ticket performance right after pruning (left) and after training (right). The baseline ticket performance is indicated by the black dashed line, leftmost sparsity correspond to planted ticket sparsity.

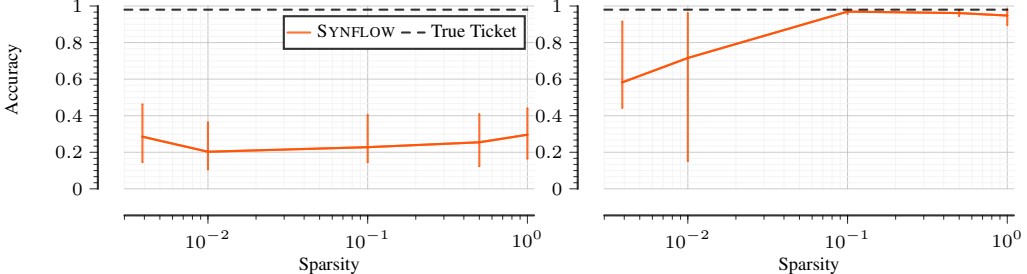

Figure 13: *Singleshot results for* SYNFLOW *with* 100 *pruning rounds.* Performance on test data are plotted for `Circle` against target sparsities. We report mean and obtained intervals (minimum and maximum) across 10 repetitions of ticket performance right after pruning (left) and after training (right). The original network is of depth 6 and width 100. The baseline ticket performance is indicated by the black dashed line, leftmost sparsity corresponds to planted ticket sparsity.

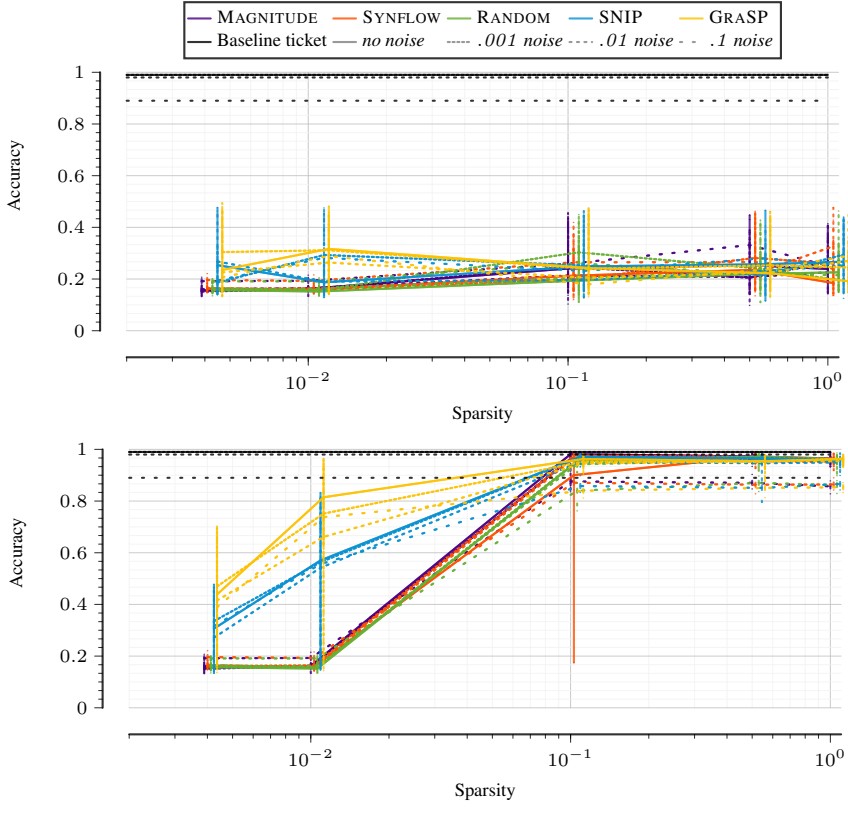

Figure 14: *Varying noise.* Performance of methods for `Circle` with varying noise for 10 rounds of alternating pruning and training. We report mean and obtained intervals (minimum and maximum) of accuracies of the final pruned network across 10 repetitions before (left) and after (right) the final training. The noise level is indicated by line type. Baseline ticket accuracy is given in black.

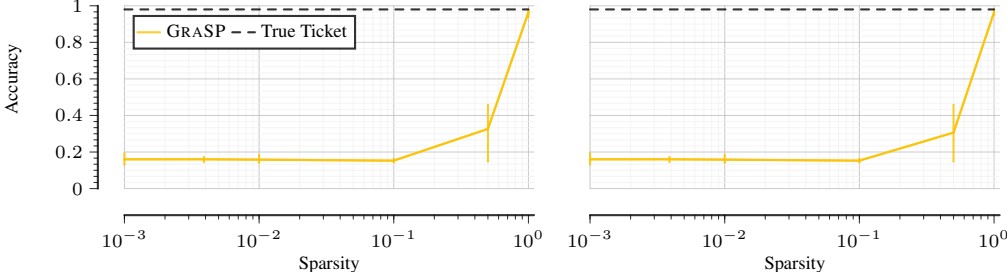

Figure 15: *Multishot with local pruning.* Performance on test data are plotted for `Circle` against target sparsities. We report mean and obtained intervals (minimum and maximum) across 10 repetitions of performance after pruning (left) and after training (right). Baseline ticket performance is indicated by the black line, second to left sparsity correspond to planted ticket sparsity.

