# OpenReview forum: "Plant 'n' Seek: Can You Find the Winning Ticket?"
_ICLR.cc/2022/Conference — ICLR 2022 Poster_

### Official Review · Reviewer_RZKn · 2021-10-30

**Correctness:** 3
**Technical Novelty And Significance:** 2
**Empirical Novelty And Significance:** 2
**Recommendation:** 5
**Confidence:** 3

**Main Review:**

##########################################################################

Pros:

(1) The paper gives a lower bound of strong winning tickets with the same depth as the target network, which reduces the dependence on the larger model depth.

(2) The winning ticket planting is quite interesting and can provide ground truth for pruning before training.

##########################################################################

Cons:

(1) I miss the motivation of planting the winning ticket to the large network. How was the target network selected? Will this winning ticket still win after being planted into the larger network? A further experiment to validate the effectiveness of winning ticket planting is required. What's more, as shown in "Stabilizing the Lottery Ticket Hypothesis", the initialization might not be enough to guarantee the matching performance. If this planted ticket can not match the full accuracy, it is reasonable that various pruning methods e.g., SNIP, GraSP, achieve unsatisfied performance.

(2) We are missing the experiments without ticket planting. Will the accuracy of various pruning methods remain similar or not? This will help us understand the role of ticket planting better.

(3) It's not clear the novelty of this work to me. The empirical findings in the paper seem already be presented in the previous work Frankle2021, which includes larger architecture and datasets. Is the bound provided tighter than the existing works?

(4) How are neurons matched in the lottery ticket planting? I suppose that the matching between the target ticket after training and the initial network is difficult.

Minor typos:

(5) Most quotation marks in the paper are used incorrectly, e.g., ’winning tickets’.

## After rebuttal

Thanks a lot for the response!

After reading the response, the motivation and contribution are clearer to me. However, I am not convinced about the effectiveness of ticket planting. I believe more elaborate sentences/figures/diagrams are necessary to help explain the whole process of ticket planting, as the core contribution of the paper. I notice that I am not the only one who is confused by this planting process. I encourage the authors to add the empirical results to support the effectiveness of ticket planting as well. Overall, I decided to increase my grading to 5.

**Summary Of The Paper:**

This paper develops a framework that allows planting and hiding winning tickets within a randomly initialized network and can be further used to assess the state-of-the-art pruning before training methods. Their empirical results of three common challenges are in line with the previous findings in Frankle2021.

**Summary Of The Review:**

The motivation and the correctness of the lottery ticket planting are not clear to me. And the empirical findings in this paper are already presented in the previous paper Frankle2021.

---

> ### Author Response · Authors · 2021-11-18
> **We have clarified the motivation behind planting and the novelty of our our contributions.**
>
> We thank Reviewer RZKn for the constructive feedback. We address the raised con points below and have updated our revised manuscript accordingly.
> 1) Motivation of planting: Planting ensures the availability of an optimal solution for pruning algorithms, i.e., a ground truth ticket that achieves optimal generalization performance and is of extreme sparsity in our experiments. The answer is therefore, yes, the winning ticket will still win after being planted.
> Comparisons with a ground truth define the gold standard to evaluate the performance of new methods in Statistics, Machine Learning, etc. They are equally important as evaluations on real data when the assumptions made during method development are not necessarily met anymore. For Lottery ticket pruning specifically, they are important to assess whether pruning algorithms are able to identify extremely sparse tickets. This is a relevant question because the theory on LT existence does not make claims about highly sparse tickets and also state-of-the-art LT pruning algorithms have difficulties to identify highly sparse tickets as subnetworks of randomly initialized neural networks. For that reason, some papers propose to resort to rewinding tickets (see "Stabilizing the Lottery Ticket Hypothesis"), fine-tuning, etc., essentially abandoning the quest for Lottery Tickets as subnetworks of randomly initialized neural networks.
> We show with our planting experiments that the challenges of sparse ticket identification are likely of algorithmic rather than fundamental nature. Possibly, rewinding, etc., are not necessary if the underlying pruning algorithm is improved.
> 2) The methods perform the same without planted solutions. The planting does not seem to affect the performance of the methods, as they search for different architectures than the planted ones (and find less performant tickets at low sparsity levels). We have added a comment on the observation, since this should indeed be mentioned. It also indicates that our planting is successful in the sense that it does not destroy the ability of pruning algorithms to find tickets.
> 3) Novelty of findings:
> Frankle et al. cannot assess whether the analyzed pruning algorithms can find extremely sparse tickets (because they have no ground truth available). In fact, the authors themselves discuss the limitations of their analysis and criticize the lack of comparisons with a ground truth, which we provide.
> Furthermore, we do not simply confirm their findings. We find that an iterative extension of synflow outperforms Iterative Magnitude Pruning in several cases.
> In addition, we compare with edge-popup and compare all algorithms’ ability to find strong lottery tickets.
> We have also added comparisons with dynamic sparse training and fine-tuning to the supplement.
> Finally, we highlight a current limitation of state-of-the-art pruning algorithms to find extremely sparse tickets. This limitation is likely of algorithmic and not fundamental nature. In consequence, extremely sparse solutions to complex learning tasks might exist as subnetworks of randomly initialized neural networks. We are just not able to identify them yet with the available algorithms.
> We could only come to this conclusion with our experiments based on planted ground truth tickets.
> Furthermore, please note that our contribution is not purely experimental. We also derive a planting framework and ground truth tickets.
> 4) The matching is defined as explained in Sec. 2.2 and A.3. We assume that the target network is given (not necessarily obtained by training but it could be as e.g. in case of the VGG-16 results). We then plant the target by adjusting a subnetwork of a randomly initialized neural network.
> To reduce the change that we have to apply we try to match neurons in the target network with neurons in the mother network. Essentially, we search for a neuron that has parameters which are closest in L2 norm to the parameters of a target neuron (respecting scaling).
> 5) We have corrected the quotation marks.

---

> > ### Comment · Reviewer_RZKn · 2021-11-26
> > **Thank you for the response**
> >
> > Thanks for your clarifications on the motivation and contribution. The motivation for planting is very clear to me and I believe such planting is important to identify if the current limitation of pruning methods is algorithmic and or fundamental.
> >
> > However, I am still confused about the effectiveness of the planted target ticket. Instead of saying "The answer is therefore, yes, the winning ticket will still win after being planted.", could the authors directly show the accuracy achieved solely by the planted ticket? For instance, in Figure 3 and Figure 4, does "baseline ticket" refer to the target ticket before planting or after planting? If it is the latter case, is this accuracy achieved solely by the planted ticket without involving let's say, other non-scaled parameters or this is the accuracy achieved by the dense networks including the planted ticket?

---

> > > ### Author Response · Authors · 2021-11-26
> > > **Clarification on target ticket accuracy**
> > >
> > > We thank Reviewer RZKn for giving us the opportunity to clarify this point. The accuracy of the baseline ticket in Figures 3 and 4 corresponds to the accuracy (or mse) of the target ticket before planting. It also corresponds to the target ticket after planting if we set all other parameters in the mother network that do not belong to the ticket to zero and apply the right scaling factor to the output. Note that we also estimate and apply the best scaling factor to the results of all pruning algorithms. Our comparison is thus fair.
> > > The dense mother network does not achieve a good accuracy at initialization in general (with our without planted ticket). This is why we need training or pruning or both.

---

> ### Author Response · Authors · 2021-11-29
> **Our planted tickets perform perfectly.**
>
> We thank Reviewer RZKn for the update and the score increase but wonder about the reasons for his/her recommendation of rejection.
> We want to highlight again that a ground truth ticket (the subnetwork) performs perfectly (as indicated by the black line in the figures) both before and after planting, in fact, as good as a trained dense network. No other reviewer was confused about this fact.
> Does anything else regarding the planting process remain unclear? The rebuttal phase was meant to clarify any misunderstandings of this form and we believe we have addressed all of them. Furthermore, we have explained the process of ticket planting in detail in the supplement and also explained the main ideas in the main manuscript.

---

### Official Review · Reviewer_HJLz · 2021-11-03

**Correctness:** 4
**Technical Novelty And Significance:** 3
**Empirical Novelty And Significance:** 4
**Recommendation:** 6
**Confidence:** 3

**Main Review:**

Strengths:
- Well written paper overall, with clear motivation, experimental method and interesting results/conclusions.
- Proposed idea is, as far as I'm aware, novel when applied to sparse neural networks.
- The experimental setup is overall very well thought out (save the potential issues below), and interesting. It is complementary to the many papers that explore this topic in real models/datasets empirically.
- The paper presents a novel take on the ability to find good sparse subnetworks before and after training, with the results suggesting that the state-of-the-art methods for finding "strong" (before training) subnetworks do not perform as well as those for finding "weak" subnetworks even in for these toy problems — results matching the conclusion of other recent works in the field, but from a very different angle.
- Common wisdom for the lack of good results at high sparsity is, I believe, that at extremely high sparsity, there are just not enough weights to solve the problem. However this work shows that in cases where we know there are extremely sparse solutions, and we have enough weights, these solutions are still not found; it seems in part due to layer collapse. This result is actionable, and provides an interesting research direction for improving sparsity in neural networks, can we find methods that better avoid mode collapse and are able to discover these "planted" tickets, and does this generalize to real-world problems?

Weaknesses:
- The abstract, unlike the rest of the paper, is relatively poorly written and does not do the paper justice as it is. For example, "for which we lack ground truth information" is very ambiguous, this should read, "for which we lack knowledge of ground-truth solution subnetworks" or something similar. I would suggest the authors replace this with an abstract based on the first paragraph of their conclusion, which presents a much better summary of the paper than the current abstract.
- This work shares some similarity with "Large Automatic Learning, rule extraction and generalization", Denker et al., 1987. In that work small neural networks were trained on the toy binary problem of identifying binary sequences as either having "one" or "two or more" clumps/clusters. The solutions were compared with ground-truth human proposed solutions much like in this work. *However* that work showed that dense NNs do not necessarily identify weights/structure (i.e. subnetworks) that humans would identify/hand design, but they do find equally performant solutions (that weren't as compact) — an important finding that seems relevant to this work, and suggests an important, but missing issue with the author's experimental setup: **Can dense NN find these hand-designed tickets/solutions?** Perhaps this is not a restriction of sparse DNN search/training algorithms, but a problem with NN training in general?
- I would have liked to have seen the result from dynamic sparse training methods for finding weak tickets (such as RiGL), which are state-of-the-art in (weak) post-training sparsity. In particular the ability of DST methods to potentially avoid layer collapse would make them an intestine avenue to being explored (just as the authors explored iterative methods). This is a major weakness in the evaluation in my opinion.
- It's unclear how much the results on these toy problems have bearing on the issues in finding subnets for real problems/datasets, the authors try to address this concern, but not enough that the reader can discount the issue — I found the experimental setup for planting subnetworks found from real models into randomly initialized models less convincing however.
- The authors should also cite previous work looking at training NNs on toy problems where the solutions are known (not necessarily the above reference, it's just the first that came to mind).
- There is very little space in the main paper dedicated to the results, with the results only being presented in 3 figures that are relatively small, although there are a lot more results in the appendices.


**Summary Of The Paper:**

The authors identify that contemporary methods for finding very sparse subnetworks in deep neural networks (DNNs), including both methods for finding either "weak tickets" (after training) or "strong tickets" (before training), do not find very sparse (and good) solutions. They question if this is a fundamental limitation (i.e. these very sparse solutions don't exist), or if this is a limitation of current methods in finding such solutions. To answer this question, the authors propose to either "plant" known good very sparse subnetworks in DNNs, or to try and find solutions for toy problems for which the authors could identify good hand-made very sparse subnetworks - including a classification problem (Circle), function regression (ReLU), and manifold-learning problem (Helix) . The experimental results analyze the performance of strong and weak sparse subnetwork search methods, including GraSP, SNIP, SYNFLOW, alongside magnitude and random pruning in finding these planted tickets. None of the strong ticket methods are able to find the "planted" tickets for any sparsity level, while the weak methods are able to find tickets, but not at extreme sparsity. Further analysis reveals layer collapse in particular to be a problem. The authors conclude that rather than a fundamental limitation, current methods (in particular for strong tickets) are limited in finding very sparse tickets even if they are known to exist at that sparsity level.

**Summary Of The Review:**

This is overall a very well written paper (strangely with the exception of the abstract), with a clear motivation and experimental method that I believe is novel in the sparse NN realm. The paper asks fundamental questions on the ability of contemporary methods for finding sparse subnetworks both before training, with an interesting experimental setup, and finds results (in the toy problem setting) that suggest these methods are lacking. Layer collapse specifically is identified as one of the problems leading to this issue, and the paper is somewhat convincing that the problem of finding very sparse solutions is algorithmic rather than a fundamental limitation.

Unfortunately the paper loses some relevance in that it lacks any analysis of dynamic sparse training methods in addressing this problem, also it's not clear that the author's implicit assumption that a dense DNN could find these hand-designed solutions is true — perhaps this is not a restriction of sparse DNN methods but NNs in general. Nevertheless, overall it remains interesting and relevant overall, and I recommend its acceptance - although I encourage the authors to revise the abstract.

---

> ### Author Response · Authors · 2021-11-18
> **We have added a comparison with RiGL.**
>
> We thank Reviewer HJLz for the in depth review and the positive assessment of our work. We address the points of weakness in our revised manuscript and respond in detail below.
> 1) We have revised the abstract as suggested.
> 2) We thank Reviewer HJLz for the very interesting reference and inspiring topical connection. We are happy to discuss the paper together with additional literature of a similar spirit in our revision. These works also identify great candidates for planting. Some of the papers compare directly with solutions that dense neural networks learn and find high discrepancy between learned at hand-designed architectures. Sometimes, NNs even fail to learn solutions that are competitive with the hand-designed ones.
> 3) Can dense NN find our hand-designed tickets/solutions? - Yes and no. First, we would like to highlight our experimental results at sparsity level 1, which are dense networks. All our proposed problems are solvable by dense networks in the sense that they learn models that achieve the same performance as the ground truth ticket. These dense solutions are yet not identical to the ground truth tickets, as they are not sparse (i.e., they do not learn to set most parameters to zero). This is also in line with the literature, as suggested by Reviewer HJLz. In contrast, pruning of a NN tries to address this issue by searching for sparse solutions. Our analysis could thus be phrased in a similar spirit as the literature on training dense NNs, as we ask the question: Can pruning of NNs identify highly sparse hand-designed architectures? Even though our answer is currently no, the hope is that lottery ticket pruning in general could be able to perform efficient structure learning in the long run. Contemporary algorithms come close in case of our CIRCLE problem.
> 4) Comparison with dynamic sparse training: We thank Reviewer HJLz for pointing out RiGL, which is an exciting method that is able to prevent layer collapse. However, it does not directly fit into the scope of our analysis, because it does not identify a lottery ticket in the sense that it finds a subnetwork of a randomly initialized neural network. Thus, our planting framework might be less relevant to evaluate RiGL. However, we can still assess its ability to find extremely sparse tickets by comparison with our ground truth networks. To satisfy the curiosity of Reviewer HJLz, we have added such a comparison of sparsity levels in the supplement and discuss it shortly in the main manuscript. We find that also RiGL is not able to identify LTs of the same sparsity as our ground truth tickets - yet.

---

> > ### Comment · Reviewer_HJLz · 2021-11-23
> > **Thanks for the answers/clarifications**
> >
> > I would like to acknowledge and thank the authors for their answers and clarifications.
> >
> > While I'm somewhat with Reviewer2wJr on wanting to see more "weak" methods analyzed, I think it is fair that the authors decided to focus on "strong" methods given the appropriate motivation in the paper. I think this point could be made clearer in the paper - i.e. that the paper's focus is strong rather than weak, and that the authors are aware of the many weak methods they are not looking at (i.e. they are mentioned in the background), with a clear motivation for why this focus is justified. I personally appreciate the extra results on dynamic sparse training methods in any case, as it wasn't clear if they would perform differently or not, and I believe this will be very interesting for future research.
> >
> > I also think it would be good to bring up (briefly) this very interesting and poorly understood result we have discussed here (as the authors have said they will do) that human "planted" solutions are not typically found emperically, but rather equally good but perhaps not as sparse solutions are instead found by both sparse and dense DNN training algorithms. As the paper is currently written, it sounds like the authors expected the planted solutions to be found, rather than using them as a baseline for generalization/sparsity performance. Given this usage of the 'planted' solutions as a baseline, it also brings the question to mind of if these 'planted' solutions are in fact provably lower-bounds on the sparsity of a 'good' solution for these relatively simple problems, or just the most 'intuitive' solutions that happen to be very sparse?

---

> > > ### Author Response · Authors · 2021-11-25
> > > **We agree with Reviewer HJLz**
> > >
> > > We thank Reviewer HJLz for the constructive comment and would be happy to emphasize the fact that we plant strong lottery tickets. The motivation to do so is that we guarantee solutions for pruning algorithms that identify lottery tickets as subnetworks of randomly initialized neural networks. These could be identified by strong or weak pruning algorithms alike, but they do not fulfill this role for related approaches like DST or fine-tuning.
> > >
> > > Planting is stronger than just providing a benchmark, as it guarantees solutions. However, our ground truth tickets could also serve as benchmarks to a wider class of algorithms. For that reason, we thank Reviewer HJLz and vW7Q for the great idea to include experiments on RiGL and fine-tuning, as these outline exciting future avenues for our framework.
> > >
> > > Are the considered approaches able to recover the planted tickets or, more generally, are these approaches finding similar solutions to human constructed ones? We would be happy to discuss this question in more depth. In our analysis, we do not require pruning algorithms to identify our planted tickets exactly but only tickets of comparable sparsity and performance. All algorithms are allowed to find non-human constructed solutions and even sparser ones than our baselines if these exist.
> > >
> > > Do we expect that LT pruning algorithms will be able to meet the sparsity of our ground truth tickets? - In fact, for CIRCLE, i.e. our classification problem, the iterative ‘weak’ methods come close to solving it, with almost as performant tickets as the planted baseline with sparsity equal to the baseline. It turns out that the studied methods struggle more with regression tasks. This observation might be explained by the fact that most lottery ticket pruning algorithms are currently developed and evaluated solely in the context of classification. By raising awareness, we hope that over time pruning algorithms will also become better in regression.
> > >
> > > We consider the ReLU problem to become a relevant indicator to measure progress in this direction because our ground truth is provably the sparsest neural network representation that solves this regression problem.
> > > The other two problems have more of a baseline character. While pruning algorithms cannot find sparser solutions yet, we cannot exclude the possibility of their existence. However, in case that pruning algorithms find sparser solutions, we will likely also gain some interesting theoretical insight into how NNs model particular problems.

---

### Official Review · Reviewer_vW7Q · 2021-11-03

**Correctness:** 4
**Technical Novelty And Significance:** 2
**Empirical Novelty And Significance:** 2
**Recommendation:** 6
**Confidence:** 3

**Main Review:**

**Strength:**
1. This paper provides a framework for benchmarking different pruning methods on their abilities to identify strong/weak lottery tickets, which can provide rich insights for the lottery ticket community.
2. The proposed framework is driven by theoretical analysis.

**Weakness:**
1. One major concern is what's the practical usage of the proposed framework in real-world tasks like image classification, considering the optimal solution is no more available. Can the framework still plant near-optimal solutions and provide useful insights in these cases? Although the paper analyzes some general trends, it's not clear whether such observations can be consistently scaled up to large-scale datasets.
2. The paper is not well-written and the logical flow can be improved. For example, it's not clear the "lottery ticket" in Sec. 2 denotes strong or weak lottery tickets, leading to many confusions. The wording in Sec. 2.1/2.2 can be improved with more structural logic. In addition, the title of Sec. 2 is not accurate as "existence of lottery tickets" has been discussed in the first lottery ticket work.
3. Other questions:
- For the problems proposed in Sec. 2.3, are there any references? It's not clear why they can "reflect typical machine learning algorithms" as described in the contributions.
- It's a common practice of existing pruning methods to further fine-tune the pruned networks with inherited weights. In addition, a recent work [1] also shows that fine-tuning the identified tickets can achieve better results than retrained lottery tickets or rewinded lottery tickets, which also aligns with the common practice in network pruning. Can the proposed framework indicate what's the best pruning algorithm with the best final accuracy beyond the scope of lottery tickets?

[1] "Lottery Ticket Preserves Weight Correlation: Is it Desirable or Not?", N. Liu et al., ICML'21.

**Summary Of The Paper:**

This paper proves the existence of strong lottery tickets and further develops a framework to plant and hide winning lottery tickets with desirable properties in randomly initialized networks to help analyze the ability of state-of-the-art pruning methods for identifying tickets of extreme sparsity.

**Summary Of The Review:**

Given the concerns about the practical usage of the proposed framework, I tend to deem this paper marginally below the acceptance threshold. I'm willing to adjust my scores if the concerns are addressed.

---

> ### Author Response · Authors · 2021-11-18
> **We have clarified the logical flow.**
>
> We thank Reviewer vW7Q for the detailed feedback. Below we address the identified weaknesses.
> 1) If Reviewer vW7Q means with “practical usage” the development or improvement of a pruning algorithm, we should mention that we have indeed improved the edge-popup algorithm to find sparser strong lottery tickets. However, this is not the main contribution of our work.
> Regarding image classification, we have demonstrated that our framework is useful to benchmark pruning algorithms for strong lottery tickets by planting sparse solutions. Even though these solutions might not be optimal, they can still present a challenge for strong lottery ticket pruning.
> Also note that the relevant scaling question in our case is not the size of the dataset but the size of the initial neural network (and the target network). If we do not need to find the best neuron match (as described in the manuscript), planting has no scaling issues (see also the definition of our planting algorithm).
> Another insight of practical relevance that we deduce based on our framework is that contemporary lottery ticket pruning algorithms do not achieve satisfactory performance on simple, small scale problems. Exactly because our proposed problems are small scale, we have a chance to understand what goes wrong. We hope that this can inspire algorithmic improvements in the future.
> 2) We have added the adjective “strong” in situations of ambiguity in the revised manuscript. Also note that every strong lottery ticket is also a weak lottery ticket. Furthermore, we have discussed the theoretical literature in detail and made clear that we do not present the first existence proof on lottery tickets but only the first that proves the existence in neural networks of the same depth as the target network.
> 3.1) In our revised manuscript, we have added references that support our point that problems with ground truth are representative of standard machine learning problems: a classification problem (circle), function regression (ReLU), and a manifold-learning problem (helix). Furthermore, ReLU can be solved by the sparsest possible architecture, which is quite relevant considering that we want to study extreme sparsity. It is also a building block of more complicated target networks. Circle requires learning of the l2 norm, a building block of any radial symmetric function. Manifold learning occurs frequently in unsupervised learning problems.
> 3.2) Fine-tuning: We are familiar with the indeed exciting work [1]. However, it is not directly relevant for our work, which focuses on lottery tickets that are subnetworks of randomly initialized neural networks and thus pruning algorithms that identify such tickets. We do not see what kind of related research questions could be answered with the help of planting. However, our identification of ground truth tickets could still be relevant for a comparison of sparsity levels. To satisfy the curiosity of Reviewer vW7Q, we have added experiments in the supplement that show that also fine-tuning does not reach the sparsity level of our ground truth tickets.

---

### Official Review · Reviewer_2wJr · 2021-11-04

**Correctness:** 3
**Technical Novelty And Significance:** 2
**Empirical Novelty And Significance:** 2
**Recommendation:** 5
**Confidence:** 3

**Main Review:**

I think the idea of embedding hidden tickets inside a network to evaluate lottery ticket hypothesis is interesting, but this paper is let down by the mismatch between the proposed method and experiments section:

The derived lower bound applies to strong lottery ticket hypothesis but the experiments section mostly focuses on pruning-at-initializaton methods such as SNIP, GraSP, and SynFlow that were clearly designed for a different (e.g. weak LTH) setup. Given that one of the claimed aims of this paper is to analyze the ability of state-of-the-art pruning methods in finding tickets, the authors should elaborate on how the insights from their approach are transferable to these methods. The authors do note this gap but claim that the insights are similar to known trends in image classification and therefore the insights are transferable. However, I'm not sure how this setup can be used to make any new claims about pruning-at-init methods.

I'm also not convinced about the three designed tasks in the experiments section. The authors claim they construct tickets for tasks that represent "typical machine learning problems" but these tasks are not typically used in the LTH literature. It would be great if authors discuss how these are relevant to the image classification task that LTH literature mainly targets. Some citations to where similar tasks are used would also be great.

### Lower bound on existence probability

- The authors prove the lower bound for MLPs but say it can also be applied to convolutions. Does this include architectures that include BatchNorm and/or residual connections. Given that these are the most commonly used architectures, the authors should discuss in more detail how the proof applies to these models.
- I think it would be great if authors could provide plots that give better intuition for theorem 2 in terms of what kind sparsities one can expect for typical architectures. Perhaps as a function of in-degree and width of the parent network? Basically it's not immediately obvious to me how tight these bounds are.
- The authors exclude the last layer from the strong lottery ticket hypothesis and mention it is common practice. Could you cite some of the references where this is common? I believe the original paper by Ramanujan did include the last layer (and BN layers). It would be great if the authors could also provide ablations to show how much this assumption skews the results?

### Planting the ticket
- This seems to be a core idea of the paper but I don't think it's well described in the paper. I think the authors should elaborate on the role of scaling factors in the method. I'm assuming the whole point behind them is to do the actual "hiding" part of the method but that aspect has been somewhat lost in the current version.
- I think a diagram could be very useful in explaining the ticket planting here.

**Summary Of The Paper:**

This paper argues that one reason evaluating the strong Lottery Ticket Hypothesis is difficult is the lack of ground-truth tickets. They circumvent this by embedding a winning ticket in the weights at initialization and evaluating how well different methods can recover it.

**Summary Of The Review:**

Interesting idea but there's a mismatch between the proposed theorem and the experiments.

---

> ### Author Response · Authors · 2021-11-18
> **We have clarified the connection between our experiments and theory in the revised manuscript.**
>
> We thank Reviewer 2wJr for the detailed review. We address each point of critique below.
> 1) Perceived mismatch between experiments and theory:
> The main focus of our theory (as well as of any other existence proof for lottery tickets in the literature) are strong lottery tickets. Any pruning algorithm that identifies a subnetwork of a randomly initialized neural network (i.e. a lottery ticket) could in principle find this ticket. Further training of the ticket, whose existence we have proven or guaranteed by planting should not reduce the performance of the initial ticket. (Otherwise, the training algorithm should be reconsidered.) Note that any strong lottery ticket is also a weak lottery ticket. Thus, any lottery ticket pruning algorithm that searches for a weak ticket could, theoretically, also find a strong lottery ticket. In principle, it could be the case that a strong lottery ticket does not exist and therefore pruning algorithms have to resort to finding weak tickets. Yet, if a strong ticket does exist, also a pruning algorithm for weak lottery tickets could use the strong ticket and the fact that the strong ticket exists should then also improve the performance of the weak lottery ticket pruning algorithm. In fact, one could argue that an excellent pruning algorithm should find the sparse strong ticket, because this saves a lot of computations later as it makes further training unnecessary.
> However, our experiments show that weak pruning algorithms really do something different and that they generally do not find strong tickets, even when strong tickets exist and even if the algorithms are allowed to train iteratively (see for instance Fig. 5 left column).
> Furthermore, edge-popup, the algorithm that explicitly searches for strong lottery tickets, cannot find the planted ticket of extreme sparsity. Regardless, the other algorithms find tickets that, before training, cannot compete with the tickets found by edge-popup. Thus, weak lottery ticket pruning algorithms neither find initial (strong) tickets that are competitive with the target ticket nor with the strong tickets found by edge-popup.
> These insights could only be gained by planting strong lottery tickets, as this ensures their availability to all algorithms. We therefore do not see a mismatch between our experiments and theory. However, we agree that we should have clarified this line of thought as done now in our revision.

---

> > ### Author Response · Authors · 2021-11-18
> > **Planting ground truth tickets is necessary to answer our research questions.**
> >
> > 2) Lottery tickets (LTs) have not only been used to solve image classification tasks but also for natural language processing, reinforcement learning, on tabular data, etc. Finding sparse neural network architectures is relevant really wherever large, overparameterized neural networks are in use. They are particularly important when deep learning is our only (or at least best) option to solve a specific task (like image classification). If we could design competitive hand designed neural network architectures to solve these tasks, probably, we would not need deep learning anymore. Thus, a restriction to image classification problems would likely mean abandoning any hope of comparisons with a ground truth. Yet, we could not answer the questions that we have posed in our work without this ground truth.
> > 3) Our small scale examples are representative of standard machine learning problems: a classification problem (circle), function regression (ReLU), and a manifold-learning problem (helix), which are standard problems that are discussed in any ML textbook. If we fail on these standard tasks, how can there be any hope to solve a more complex task? Furthermore, ReLU can be solved by the sparsest possible architecture, which is quite relevant considering that we want to study extreme sparsity. It is also a building block of more complicated target networks. Circle requires learning of the l2 norm, a building block of any radial symmetric function. Manifold learning occurs frequently in unsupervised learning problems. We have added references that support our point as requested.
> > 4) We have not claimed that our lower bound applies to convolutions, only our planting algorithm does. That said, we could extend our proof to convolutional architectures and skip connections, as long as the large initial network has enough channels or skip connections to prune from. The limiting factor would again be degrees of the neurons or filter sizes of the target network. The argument is exactly the same as for the presented fully connected networks. For simplicity, we therefore skip a detailed theoretical treatment of convolutions and skip connections. Batch norm parameters can be integrated into the weights and biases of a trained network and thus into the target network. They do not require special consideration for pruning. They only help with training the parameters of a network.
> > 5) We have added plots to the supplement that show how our bound on the existence probability depends on various parameters of the target and mother network architecture. Unfortunately, this cannot tell us how tight our bounds are. (To answer this question we would need to be able to compute the actual existence probability or be able to find tickets with high probability if they exist by chance.) But we can give an intuition about what kind of architectures could exist as lottery tickets. Furthermore, we have computed the bound for our ground truth tickets, as this strengthens our motivation to plant.
> > 6) We do not exclude the last layer from strong ticket pruning. This was a misunderstanding. We simply remarked that the last layer presents a challenge for pruning. This challenge could be overcome by a really wide second to last layer, or by considering an initial network of depth L+1, or by allowing to train the last layer. We do not need to do any of that because we plant a good ground truth.
> > We have removed this confusing remark in our manuscript.
> > 7) Unfortunately, we have to work with a space constraint. We did our best to better explain the role of scaling factors in the revised manuscript.

---

> > > ### Comment · Reviewer_2wJr · 2021-11-29
> > > **thanks for your reply**
> > >
> > > I'd like to thank the authors for their rebuttal. I particularly appreciate the summary of changes to the paper. I currently remain unconvinced that my concerns have been addressed to the point where I'd increase my score, however, I'll read the new revision in its entirety again and will keep an open-mind when discussing the paper with other reviewers and the ACs.

---

> > > > ### Author Response · Authors · 2021-11-29
> > > > **What is missing?**
> > > >
> > > > We thank Reviewer 2wJr for the update and kindly ask to clarify why our rebuttal does not convince him/her?  In particular, what are the specific details that are left to explain and why do our answers not address those properly?
> > > > Note that the discussion period ends today. Now is the last opportunity to read the updated draft and form an opinion. There is no time for further discussions with the AC or other reviewers.

---

### Official Review · Reviewer_kf7j · 2021-11-09

**Correctness:** 4
**Technical Novelty And Significance:** 3
**Empirical Novelty And Significance:** 3
**Recommendation:** 6
**Confidence:** 4

**Details Of Ethics Concerns:**

There are no ethics concerns.

**Main Review:**

The authors provide an interesting new avenue for evaluating pruning methods, with the use of ground truth sparse networks.  This contribution relies on a theoretical result, which allows us to place a lower bound on the probability of a strong, sparse network.  These both are useful contributions.

From a practical, empirical perspective, I do have concerns about the impact of this result, as the authors note that strong networks are currently still difficult to find.  Moreover, I also have some concerns that the outcome of using ground-truth strong networks largely confirms prior work (Frankle et al., 2021, Ramanujan et al. 2020) and yet relies on much smaller networks for evaluation than the networks evaluated in prior work.  The VGG-16 task is presented as a bit of an afterthought and is not central to the experiments, so I am uncertain how this planting algorithm can be scaled to similarly larger tasks. At the same time, I share the author's hope that using ground truth strong networks will spur new methods for the finding of strong, sparse networks.

My suggestions for the paper are mostly cosmetic in nature.  First, the inclusion of Zhou et al 2019, would provide a fuller picture of the history of research into strong lottery tickets, as Ramanujan et al. 2020 write that their work was inspired by the results shown in Zhou et al 2019.  Additionally, the description of "confidence" in the plots is not only vague, but not visible based on the format of the plot markers.  These markers are too large for the current "confidence" metric, which is not explicitly described in the paper.  Organizationally, information from A2, A3, and B1 were most helpful to me personally in understanding the paper.   In fact, I found section A3 and B1 more useful to understand the experiments than sections 2.2 and 2.3. I believe that content from these sections would better help the reader if they were presented in the main sections of the paper.  Whitespace used in the main paper could be reorganized, and figures 1 and 2, I believe were less helpful in understanding the paper and could therefore be moved to the appendix.

----

Based on the additional information provided by the authors in response to reviewer feedback, I would also like to increase my review score.


**Summary Of The Paper:**

The authors note a distinction between kinds of sparse networks in the literature. "Weak tickets" require training to perform comparably to the original network, while "strong tickets" do not.  The authors prove a lower bound for the probability that a strong ticket exists, and note that instead of training, this "strong ticket" need only be scaled by some constant to achieve similar performance to the larger network.  Using the insights of this proof, the authors propose three benchmark tasks to find a ground truth sparse network.  They evaluate common pruning methods to find both weak and strong tickets using these tasks, and find that most methods perform well on 2 out of the three weak ticket finding tasks.  They also note that the only method designed specifically for strong ticket finding performs well on these tasks.  For further evaluation, they additionally share weak and strong ticket finding results for VGG-16.




**Summary Of The Review:**

Overall, this work I believe would be a useful contribution to the community.  I think however, further streamlining of the manuscript to improve clarity would strengthen the paper.  First, citation of Zhou et al https://arxiv.org/abs/1905.01067 would strengthen the related work, as  Ramanujan et al. note the influence of the paper on their work.  Second, re-plotting the figures so that "confidence" is both clearly defined (percentile value for the confidence is not defined) and visible within the plots, would allow the reader to better interpret the plots on their own.  Third, information from Appendix A2, A3, and B1 are useful experimental details and would help the reader if they were presented in the main section of the paper instead of the appendix.  In fact, I found section A3 and B1 more useful to understand the experiments than sections 2.2 and 2.3. Whitespace used in the main paper could be reorganized, and figures 1 and 2, I believe were less helpful in understanding the paper and could therefore be moved to the appendix.

---

> ### Author Response · Authors · 2021-11-18
> **We have streamlined the manuscript.**
>
> We thank Reviewer kf7j for the insightful review and the helpful suggestions to improve the clarity of the paper.
> 1) We would like to mention that we do not simply confirm the findings of Frankle et al., which suggest that Iterative Magnitude Pruning (IMP) is the state-of-the-art pruning algorithm for (weak) lottery tickets. In contrast, we show that none of the studied algorithms (including IMP) can find tickets of the sparsity of a planted ground truth ticket. In fact, Frankle et al. themselves discuss the limitations of their analysis and criticize the lack of comparisons with a ground truth. We fill this gap by providing this ground truth. We emphasized this point in our revised draft.
> 2) Furthermore, we find that an iterative application of synflow outperforms IMP in several experiments. In addition, Frankle et al. are not considering strong lottery tickets at all. In contrast, we identify a research gap for algorithmic improvements, as we find that none of the analyzed algorithms can identify extremely sparse strong or weak lottery tickets. This is important because it suggests that extremely sparse LT might exist as subnetworks of randomly initialized neural networks also in complex learning tasks. State-of-the-art LT pruning algorithms are just not able to find them - yet. To find them we might not need to resort to rewinding or fine-tuning techniques that give up on the idea of subnetworks of randomly initialized neural networks. We come to this conclusion by careful experimental design and planting of ground truth tickets. The VGG-16 results provide this crucial insight also in the often studied setting of standard image classification benchmarks. We have highlighted this more clearly in our revision to avoid the impression that these results present only an afterthought.
> 3) We agree that Zhou et al. should be cited and discussed properly and do so  in the revised version of our paper.
> We tried to make the confidence intervals better visible by using bands in the plots and defined confidence explicitly (here we use .95 confidence intervals).
> 4) We are happy that Reviewer kf7j found our supplementary material useful and read it in detail. Sections 2.1 and 2.2 state our theoretical results and contributions and explain the foundations of the (non-trivial) planting algorithm. They are important for the logical flow of the paper. Without Figures 1 and 2, readers would have only our theoretical short description of the tasks, which are probably difficult to understand. Besides, Reviewer 2wJr would like to learn more about the ticket architectures (which are visualized in Figure 2) and not less, which is a dilemma for us. However, we added more experimental details in the main paper to make it better understandable and hope that this helps reviewer kf7j in understanding without sacrificing foundational details that other reviewers valued.

---

### Author Response · Authors · 2021-11-29
**Summary of rebuttal and contributions**

We thank all reviewers for their constructive feedback that encouraged us to improve our paper. As there is not much time left to address any concerns that might remain, we kindly ask for your response.
We are confident that we have addressed all points of critique. Concretely, we made the following changes to our manuscript:
* We have polished our abstract and main manuscript according to the reviewers’ suggestions.
* We have emphasized the motivation behind our experiments on imaging data.
* We have explained the role of scaling factors in detail.
* We have added figures in the supplement that help in understanding our theoretical bounds.
* We have added a connection to the deep learning literature that compares trained neural networks with handcrafted solutions and extended our literature discussion accordingly.
*We have motivated our planting framework in more detail and further emphasized the call and need for ground truth tickets by the community.
* To satisfy the curiosity of the reviewers, we have added comparisons with LT fine-tuning and dynamic sparse training (RigL) by using our ground truth solutions as baseline.

Accordingly, we kindly ask all reviewers to increase their score or communicate what they deem is an open point of critique.

We are confident that our work makes a valuable contribution of interest to the ICLR community, as we enable the answer of research questions that were impossible to address previously. Furthermore, we anticipate that our contributions could guide future progress regarding the lottery ticket hypothesis.
In detail, we make the following contributions:

1. **Planting and hiding lottery tickets.** We have solved a difficult open problem: We derived a framework that plants target tickets in randomly initialized neural networks to challenge state-of-the-art lottery ticket pruning algorithms.
2. **Ground truth tickets.** We hand-crafted extremely sparse neural networks, which qualify as ground truth lottery tickets. One of these solutions is provably optimal. More generally, we have designed strong baselines for sparse training and neural architecture search.
3. **Benchmarks.** By planting and hiding solutions in randomly initialized neural networks, we have challenged state-of-the-art algorithms and highlighted their current limitations. At the same time, we have provided an interpretable, small scale test environment to discover and address the reasons for their failure.
4. **Answering fundamental research questions.** Our contributions allow us to answer research questions that were impossible to address previously, as they require a known ground truth. For instance, they allowed us to answer a question that is central to the contemporary efforts of the community to develop successful pruning algorithms: We have established that failures of state-of-the-art algorithms to find highly sparse solutions can be explained by algorithmic rather than fundamental limitations.

---

### Decision · Program_Chairs · 2022-01-20

**Decision:**

Accept (Poster)

**Comment:**

This paper takes on (in my view) one of the most important questions in the lottery ticket literature today: how small are the smallest lottery tickets that exist in our neural networks? Many methods have been proposed for finding weak lottery tickets (those that require training to reach full accuracy) and strong lottery tickets (those that do not), but we have no idea how close they come to finding the smallest lottery tickets. Moreover, in many cases, we only know how to find lottery ticket subnetworks early in training rather than at initialization. Is this a fundamental limitation on the existence of lottery tickets, or is this simply a limitation of our methods for finding them? I am personally very involved in lottery ticket conversations in the literature, and I believe I can speak with some authority when I say that these are vital questions where any progress is important.

Moreover, these are exceedingly difficult research questions, and (again, in my view) the authors should be commended for taking them on. A naive approach to these questions would involve brute force search over all possible subnetworks, which is infeasible even on the smallest of toy examples, let alone the meaningful computer vision tasks where lottery ticket work typically focuses.

I am sharing all of this information to provide background for my confident recommendation to accept this paper over the many legitimate concerns expressed by reviewers and those that I saw when reading the paper in detail. Those include that:
* This paper does not solve any of these research problems in their entirety.
* It focuses on toy networks smaller than those traditionally studied in the lottery ticket literature, and it is well known that lottery ticket behavior changes in character at larger scales.
* Planting good subnetworks may be an unrealistic proxy for the kinds of subnetworks that actually emerge naturally.
* There may be multiple good subnetworks in a network, not just the one that was planted.
* The graphs are a bit hard to read.
* I find the mix of pruning methods studied, which were designed with very different goals (pruning after training, pruning before training, finding strong lottery tickets), a bit confusing.

**The bottom line:** With all of that said, in my view, the paper asks good questions and provides an initial foothold that other researchers will be able to build on as we seek more general answers. This is similar to the contributions made by Zhou et al., which started the conversation on strong lottery tickets, and potentially even Frankle & Carbin, which kicked off the lottery ticket discussion but got many things wrong. Both papers were good first attempts at solving big problems, and both were highly influential despite their flaws. Similarly, even if this submission isn't perfect in every way, this is among the most important kinds of contributions that a paper can make. For that reason, I strongly recommend acceptance under the belief that this paper will help to foster a valuable conversation in the literature.

P.S. I really, truly, strongly beg the authors to redo their graphs following the style of some of the more user-friendly lottery ticket or pruning papers they have cited (e.g., Frankle et al., 2021). The graphs in this paper were really hard to parse. Really really really hard to parse. They're too small, the y-axis is often squished, gridlines would be helpful, the lines are overlapping in ways that are difficult to distinguish because the colors blend, etc. etc. This is quite possibly the biggest impediment I see to this paper's ability to have broader influence.